# Correction of Clcn1 alternative splicing reverses muscle fiber type transition in mice with myotonic dystrophy

Ningyan Hu[1], Eunjoo Kim [1], Layal Antoury[1] & Thurman M. Wheeler[1]✉

In myotonic dystrophy type 1 (DM1), deregulated alternative splicing of the muscle chloride channel Clcn1 causes myotonia, a delayed relaxation of muscles due to repetitive action potentials. The degree of weakness in adult DM1 is associated with increased frequency of oxidative muscle fibers. However, the mechanism for glycolytic-to-oxidative fiber type transition in DM1 and its relationship to myotonia are uncertain. Here we cross two mouse models of DM1 to create a double homozygous model that features progressive functional impairment, severe myotonia, and near absence of type 2B glycolytic fibers. Intramuscular injection of an antisense oligonucleotide for targeted skipping of *Clcn1* exon 7a corrects *Clcn1* alternative splicing, increases glycolytic 2B levels to ≥ 40% frequency, reduces muscle injury, and improves fiber hypertrophy relative to treatment with a control oligo. Our results demonstrate that fiber type transitions in DM1 result from myotonia and are reversible, and support the development of *Clcn1*-targeting therapies for DM1.

Myotonic dystrophy type 1 (dystrophia myotonica; DM1) is the most common muscular dystrophy in adults. Characteristics of this multi-system disorder include myotonia, progressive weakness, cardiac conduction delays, fatigue, sleep disturbance, and gastrointestinal dysfunction[1]. DM1 is caused by an expanded CTG repeat in the 3′ UTR of the *DMPK* gene[2]. Clinical features of DM1 arise from expression of *DMPK* transcripts containing an expanded CUG (CUG[exp]) repeat that accumulate in nuclear inclusions of affected tissues[3–6]. This pathogenic RNA readily binds proteins in the muscleblind-like (MBNL) family that are required for normal regulation of alternative splicing, gene expression, transcript stability, and alternative polyadenylation, resulting in partial loss of MBNL protein function[7–11].

Myotonia is a hallmark of DM1 and refers to a stiffness of muscles that results from delayed relaxation after voluntary contraction. In DM1, deregulated alternative splicing of the muscle chloride channel *Clcn1* pre-mRNA causes myotonia[12,13]. Aberrant inclusion of Clcn1 exon 7a shifts the reading frame, creating a premature termination codon in exon 7 and a truncated, poorly functional protein. The corresponding deficit of chloride ion conductance results in hyperexcitability and involuntary persistence of muscle activity[12,14].

Skeletal muscles are composed of individual myofibers. Expression of myosin heavy chain (MyHC) isoforms serves as a convenient indicator of the structural, functional, and metabolic phenotype of muscle fibers[15]. MyHC fiber types 1 and 2 A are rich in oxidative enzymes, fatigue resistant, and specialized for continuous activity. MyHC fiber types 2X and 2B are characterized by glycolytic metabolism, fatigue quickly, and are specialized for phasic activity. In rodents, type 2X fibers also contain moderate-to-high levels of the oxidative enzyme succinic dehydrogenase and demonstrate fatigue resistance that is intermediate between 2 A and 2B fibers[16,17]. Peak mechanical power of muscle fibers increases in order from MyHC 1 < 2 A < 2X < 2B[15].

The muscle fiber type profile is determined primarily by nerve activity and can change in response to either neural or hormonal influences[15]. Intrinsic contraction of muscle fibers that occurs independently of nerve input also is associated with changes in fiber type. For example, the Arrested development of righting (Adr) mouse model of myotonia congenita develops myotonia due to homozygous loss-of-function mutation in the *Clcn1* gene and features a predominance of oxidative fibers in most muscles[18], while a chemically-induced

---

[1]Department of Neurology, Massachusetts General Hospital and Harvard Medical School, Boston, MA, USA. ✉e-mail: twheeler1@mgh.harvard.edu

experimental myotonia shifted fiber type from glycolytic to oxidative in rats[19]. In cross sectional studies of human DM, the degree of muscle weakness is associated with Type 1 oxidative fiber atrophy and a greater proportion of mechanically less powerful oxidative fibers, which may reflect fiber type transition or a preferential loss of glycolytic fibers[20,21]. In this study, we used a double homozygous model of DM1 and antisense oligonucleotide exon skipping to test the hypothesis that deregulated alternative splicing of *Clcn1* induces a glycolytic-to-oxidative fiber type transition in DM1.

## Results

### LR41;*Mbnl1*[-/-] double homozygous mouse model of DM1

The human skeletal actin-long repeat (HSA^LR) mouse model of DM1 contains an expanded CTG repeat in the 3′ UTR of a human skeletal actin (*ACTA1*) transgene. HSA^LR line LR41 features a lower transgene copy number, a lower *ACTA1*-CUG^exp RNA abundance, deregulated alternative splicing to a lesser degree, less frequent myotonia (40% of mice examined by electromyography/EMG), and milder myopathy than HSA^LR line LR20b[12,22]. The Muscleblind-like 1 knockout (*Mbnl1*[-/-]) mouse model of DM1 features a homozygous deletion of *Mbnl1* exon 3, leading to deregulated alternative splicing and myotonia to a greater degree than in LR41 or LR20b[8,23,24].

A prior comparison of LR20b and *Mbnl1*[-/-] models found that MBNL1 protein loss-of-function can explain >80% of deregulated alternative splicing but only about 50% of the transcriptional changes initiated by CUG^exp RNA[9]. To test the hypothesis that CUG^exp RNA induces pathogenic effects in muscle tissue that are independent of MBNL1 loss-of-function, we crossed the LR41 transgenic model with the *Mbnl1*[-/-] model to create the LR41;*Mbnl1*[-/-] double homozygous model of DM1 (Supplementary Fig. 1; see Methods). LR41;*Mbnl1*[-/-] double homozygous mice experience prominent, severe myotonia that can be exacerbated in the hindlimbs by briefly grasping the base of the tail (Supplementary Movie 1). By contrast, this maneuver fails to induce visible myotonia in homozygous LR41 littermates, while visible myotonia in LR20b is relatively mild (Supplementary Movie 2). To quantify functional impairment, we monitored spontaneous activity in the x-, y-, and z-planes for 30 min using an acrylic cage and infrared lasers[25] in three age groups: 1.5–2.5 months, 3.5–4.5 months, and 6–7 months. In LR;*Mbnl1*[-/-] mice, spontaneous activity was significantly lower than in age-matched LR41 littermates, and progressively worsened with age (Fig. 1a; Supplementary Fig. 1). Mean vertical breaks, the number of single rearing events with one second between each event, was similar at the youngest age, but reduced in LR;*Mbnl1*[-/-] by 31% at the middle age, and 89% lower at the oldest age, while mean vertical counts, the total counts from the z-plane sensor, was reduced in LR;*Mbnl1*[-/-] by 32%, 55%, and 95% in the youngest-to-oldest ages examined, respectively. In LR;*Mbnl1*[-/-], total distance traveled was lower by 31%, 47%, and 59% from youngest-to-oldest, while rest time was 33% longer in 1.5–2.5 month olds, more than double in the 3.5–4.5 month olds, and more than triple in the 6–7 month olds.

Next we examined whether the absence of MBNL1 protein affected the CTG repeat length as an explanation for the greater disease severity. Previously we demonstrated that the LR20b line carries ~185 CTG repeats, shows no change in muscle tissue with aging, and is unrelated to progression of fatigue and myopathy severity[25]. CTG repeat length appeared similar in LR41 mice regardless of *Mbnl1* genotype, and, in contrast to LR20b, showed two bands. The PCR amplicon sizes of ~750 and 575 base pairs correspond to ~215 and 160 CTG repeats, respectively (Supplementary Fig. 1).

A previous study found that mice with a genetic reduction of both MBNL1 and MBNL2 proteins in muscle tissue develop DM1-like features that are more severe than absence of either MBNL1 or MBNL2 protein in isolation, supporting a model that compound loss of MBNL proteins is a critical step in DM1 pathogenesis[26]. To explore a potential role for

MBNL2 protein in mediating the more severe phenotype in LR41;*Mbnl1*[-/-], we performed fluorescence in situ hybridization, immunofluorescence (FISH/IF), and quantitative imaging of quadriceps muscle. In LR41;*Mbnl1*[-/-], CUG^exp RNA nuclear foci were evident in fewer myonuclei and, when present, were less intense, as compared to LR41 littermate controls (Supplementary Fig. 2). MBNL2 protein co-localized with CUG^exp RNA in LR41;*Mbnl1*[-/-], and the overall level of MBNL2 protein appeared higher in myonuclei and interstitial cells than in LR41 or WT controls.

### Myosin heavy chain fiber-type transition in LR41;*Mbnl1*[-/-]

To determine the relationship between myotonia severity and the oxidative state of muscle fibers in DM1 mouse models, we first performed immunofluorescence analysis of gastrocnemius muscles in mice with severe (LR41;*Mbnl1*[-/-]), moderate (*Mbnl1*[-/-]), mild (LR20b), and no myotonia (WT) using antibodies specific for myosin heavy chain type 1 (MyHC 1), MyHC 2 A, and MyHC 2B. MyHC 2X fibers remained unlabeled. In LR41;*Mbnl1*[-/-], the frequency of MyHC 2 A and 2X fibers was 60% higher than in *Mbnl1*[-/-], two- to three-fold higher than in LR20b, and four-fold higher than in WT ($P < 0.01$ and <0.0001), while the frequency of MyHC 2B was three-fold lower than in *Mbnl1*[-/-], four-fold lower than in LR20b, and five-fold lower than in WT controls ($P < 0.0001$; Fig. 1c). Labeling intensity with the anti-MyHC 2 A antibody was variable in all groups, with some fibers displaying a bright signal and others an intermediate signal between the bright 2 A fibers and the unlabeled 2X fibers.

In a prior study we found that treadmill-walking exercise combined with *ACTA1*-CUG^exp transcript-targeting ASO treatment for 3 months rescued fatigue in old LR20b mice[25]. Chronic exercise has been associated with muscle fiber type transition[15]. For example, the percentage of type 2 A fibers was higher and 2B fibers lower in mice engaged in voluntary wheel running for 4 weeks as compared to mice that received no exercise[27]. To explore the possibility that the MyHC fiber type transition in DM1 mice may be reversible, we examined gastrocnemius muscles of old LR20b mice treated with a systemic *ACTA1*-CUG^exp-targeting ASO and/or a three-month exercise training program[25]. In old mice that received the ASO, the frequency of MyHC 2X was ~50% lower and 2B 25% higher than in saline-treated controls ($P < 0.05$), and the overall fiber type distribution appeared similar to WT (Supplementary Fig. 3). Our exercise training regimen had no impact on MyHC expression patterns in old LR20b mice.

### Correction of Clcn1 alternative splicing in LR41;*Mbnl1*[-/-]

To test the hypothesis that deregulated Clcn1 alternative splicing and resulting myotonia are responsible for the fiber type transition in DM1 mice, we injected tibialis anterior (TA) muscles with an ASO that induces targeted skipping of Clcn1 exon 7a[24]. In LR41;*Mbnl1*[-/-], RT-PCR analysis revealed that ASO treatment for 16–18 days reduced exon 7a inclusion by nearly 60%, as compared to contralateral muscles injected with a control oligo (invert) that has an identical but 5′–3′ inverted sequence ($P < 0.0001$; Fig. 2a, b). After treatment for 52 days, exon 7a inclusion in LR41;*Mbnl1*[-/-] was still 40% lower in Clcn1 ASO-treated than in control-treated muscles, and in LR20b appeared similar to WT ($P < 0.0001$ and <0.01, respectively).

To confirm the specificity of the Clcn1 ASO, we examined alternative splicing of other transcripts that are deregulated in DM1 patients and mouse models[8,28–30]. In LR41;*Mbnl1*[-/-] mice, splice events of transcripts *Atp2a1*, *Clasp1*, *Ttn*, and *Cacna1s* appeared similar regardless of treatment or duration (Fig. 2c). In LR20b mice, *Clasp1*, *Ttn*, and *Cacna1s* splice events appeared similar in ASO- and invert-treated muscles, while *Atp2a1* unexpectedly showed worsening of the splicing pattern in the Clcn1 ASO-treated muscles.

The RT-PCR primers used to detect *Clcn1* alternative splicing amplify a single exon 7a exclusion (345 bp) product and three exon 7a inclusion products (424, ~550, and ~850 bp)[12]. Using standard RT-

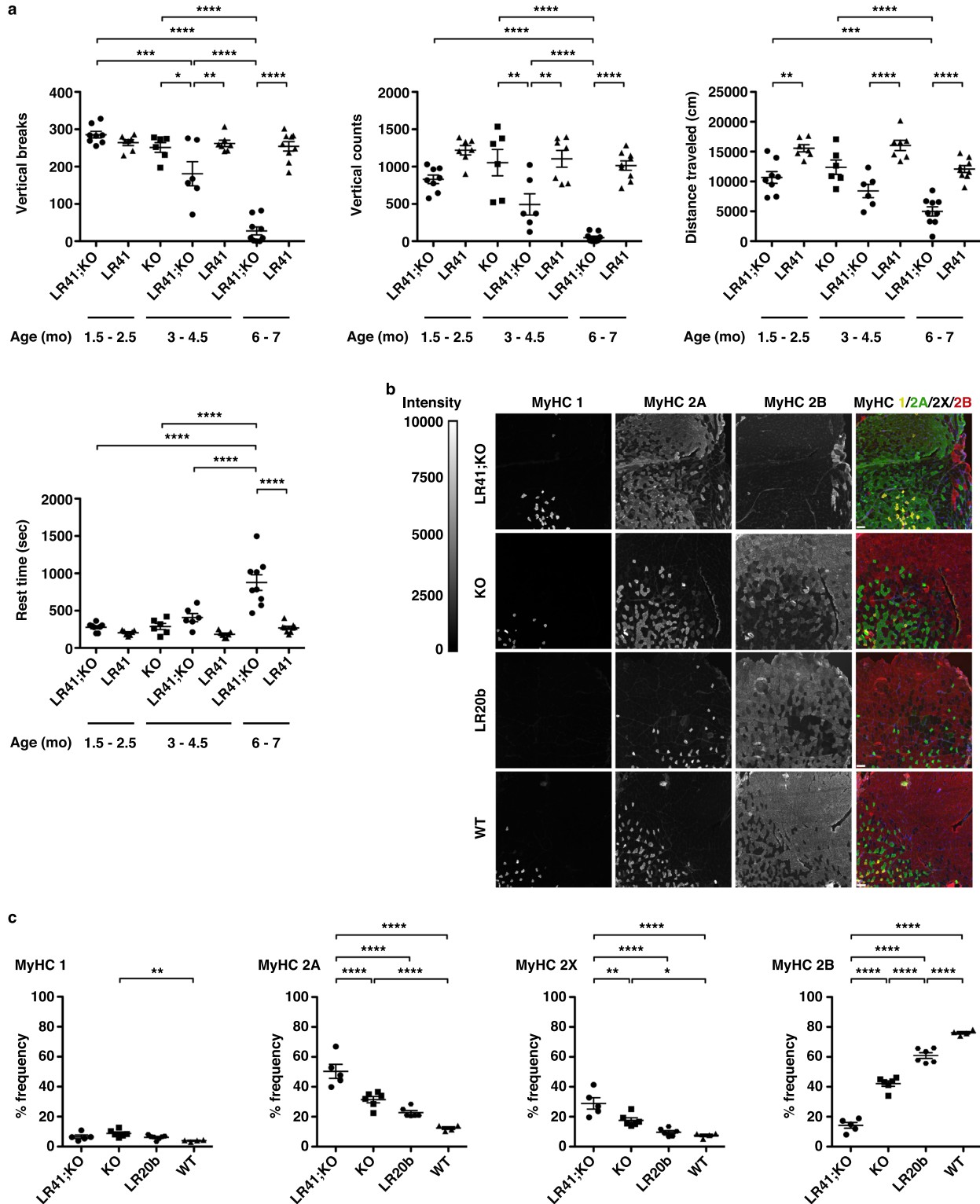

**Fig. 1 | Progressive functional impairment and myosin fiber type switch in the LR41;*Mbnl1*⁻/⁻ double homozygous mouse model of DM1.** We crossed the HSA^LR line 41 (LR41) transgenic model with the muscleblind-like-1 knockout (*Mbnl1*⁻/⁻) model to create the LR41;*Mbnl1*⁻/⁻ double homozygous model of DM1. **a** Quantification of spontaneous activity in LR41;*Mbnl1*⁻/⁻, LR41 littermates, and *Mbnl1*⁻/⁻ alone as vertical breaks, vertical counts, total distance traveled (cm), and time (seconds; sec) spent resting. We examined mice at ages 1.5–2.5 months (mo) ($N = 7$ or 8 each group), 3.5–4.5 months ($N = 6$ or 7 each group), and 6–7 months ($N = 8$ or 9 each group). The average of values for three separate 30 min monitoring sessions for each individual mouse is shown. ****$P < 0.0001$; ***$P < 0.001$; **$P < 0.01$; *$P < 0.05$ (one-way ANOVA). Error bars

indicate ±s.e.m. **b** Immunofluorescence (IF) analysis of myosin heavy chain type 1 (MyHC 1), 2 A (MyHC 2 A), and 2B (MyHC 2B) in gastrocnemius muscle tissue of untreated 2–4-month-old LR41;*Mbnl1*⁻/⁻ ($N = 5$), *Mbnl1*⁻/⁻ ($N = 6$), LR20b ($N = 6$ and WT controls ($N = 4$). Representative images of each are shown. Fluorescence intensity is 0–10,000 gray scale units. In the merge images, MyHC 1 is colored yellow, MyHC 2 A green, MyHC 2X black (unlabeled), and MyHC 2B red. Size bars indicate 20 μm. **c** IF quantification of MyHC protein expression protein expression in gastrocnemius muscles of untreated 2–4-month-old LR41;*Mbnl1*⁻/⁻ ($N = 5$), *Mbnl1*⁻/⁻ ($N = 6$), LR20b ($N = 6$), and wild-type (WT) controls ($N = 4$). ****$P < 0.0001$; **$P < 0.01$; *$P < 0.05$ (one-way ANOVA). Error bars indicate ±s.e.m. Source data are provided as a Source Data file.

PCR and band densitometry, the pre-amplification transcript stoichiometry and/or the differences in length of the PCR products may lead to overestimation of exon skipping due to preferential amplification of shorter products[31]. Alternatively, greater binding of intercalating nucleic acid dye to longer PCR products may lead to a brighter signal than in shorter products and, consequently, an underestimation of exon skipping[32]. A recent multicenter study found that droplet digital PCR (ddPCR) is more precise than RT-PCR/band densitometry for quantification of splice-shifting ASO activity[33]. To compare exon inclusion quantification methods, we developed a ddPCR assay for quantification of *Clcn1* exon 7a inclusion. Using

ddPCR, exon 7a inclusion was 50% lower in LR41;*Mbnl1*$^{-/-}$ ASO-treated muscles at 16–18 days and 28% lower at 52 days, as compared to invert oligo-treated muscles (Figs. 2d, 3; Supplementary Fig. 4). In LR41;*Mbnl1*$^{-/-}$, estimation of exon 7a inclusion by ddPCR was consistently higher than by RT-PCR, while the reverse was true in WT, although the two methods showed strong correlation (Pearson r = 0.96; $P < 0.0001$; Fig. 2e, f).

**Reversal of MyHC fiber type transition in LR41;*Mbnl1*$^{-/-}$**

To determine the effect of the Clcn1 ASO on the oxidative state of myofibers, we performed quantitative immunofluorescence for MyHC

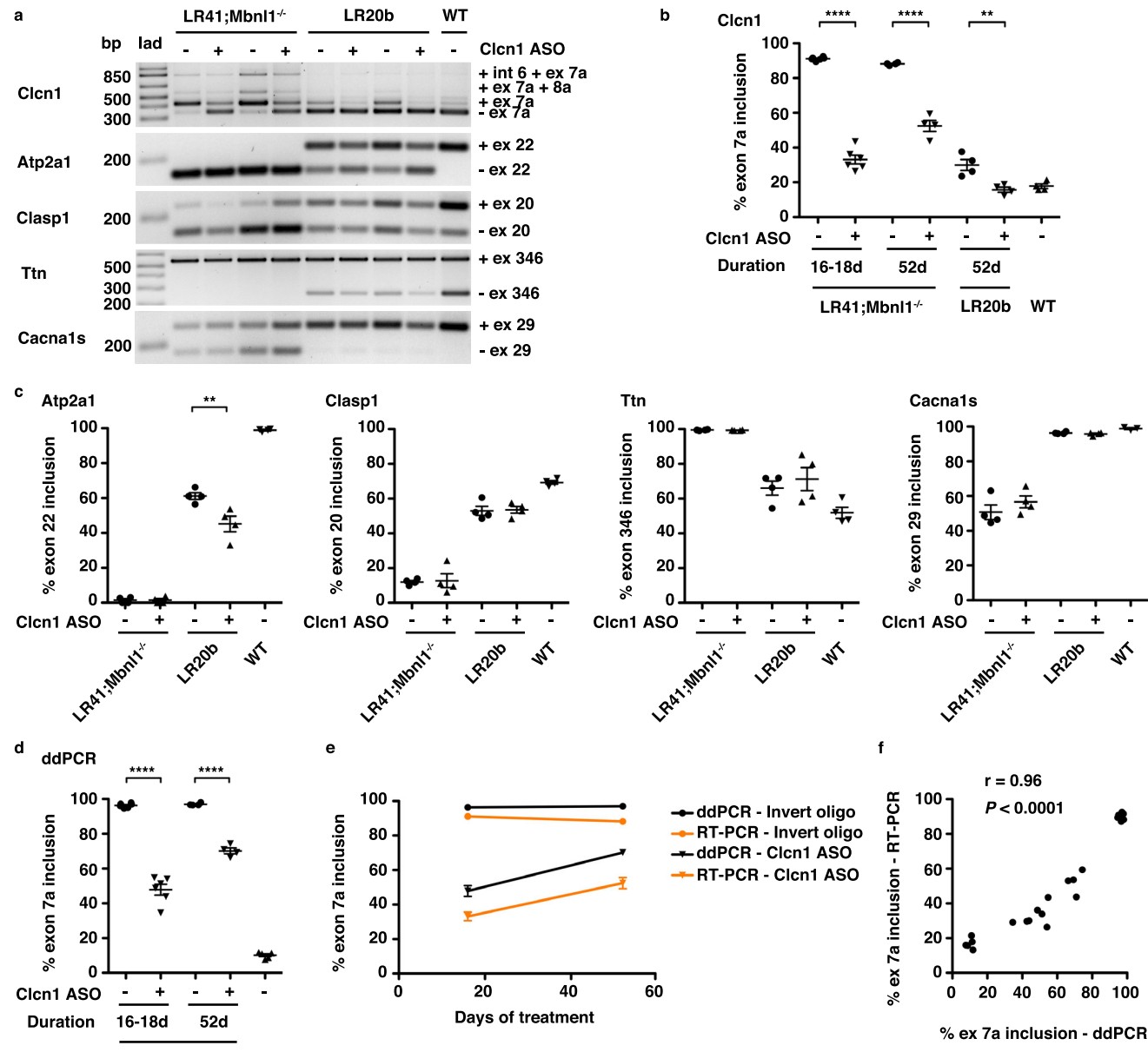

**Fig. 2 | Correction of *Clcn1* alternative splicing in ASO-treated LR41;*Mbnl1*$^{-/-}$ and LR20b mouse muscle.** We injected the tibialis anterior (TA) muscle of LR41;*Mbnl1*$^{-/-}$ (N = 10) and LR20b (N = 4) with an antisense oligonucleotide (ASO) designed to induce skipping of *Clcn1* exon 7a (+). The contralateral TA was injected with a control oligo (−) that has an identical but 5′–3′ inverted sequence and no target. Untreated wild-type (WT) mice (N = 4 or 5) served as controls. Treatment duration was 16 or 18 days (16–18d) or 52 days (52d). **a** Representative RT-PCR analysis of alternative splicing of *Clcn1* exon 7a, *Atp2a1* exon 22, *Clasp1* exon 20, *Ttn* exon 346, and *Cacna1s* exon 29 in muscles treated for 52 days. bp

base pairs, lad DNA ladder. **b** Quantification of *Clcn1* alternative splicing by RT-PCR. ****$P < 0.0001$; **$P < 0.01$; (one-way ANOVA). **c** Quantification of *Atp2a1*, *Clasp1*, *Ttn*, and *Cacna1s* alternative splicing after 52 days of treatment. **$P < 0.01$; (one-way ANOVA). **d** Droplet digital PCR (ddPCR) analysis of *Clcn1* alternative splicing. ****$P < 0.0001$; (one-way ANOVA). **e** Quantification of *Clcn1* exon 7a inclusion by ddPCR (black) and RT-PCR (orange). **f** Quantification of *Clcn1* exon 7a inclusion by ddPCR (x-axis) and RT-PCR (y-axis). The Pearson correlation coefficient r and P-value are shown. Error bars indicate ±s.e.m. Source data are provided as a Source Data file.

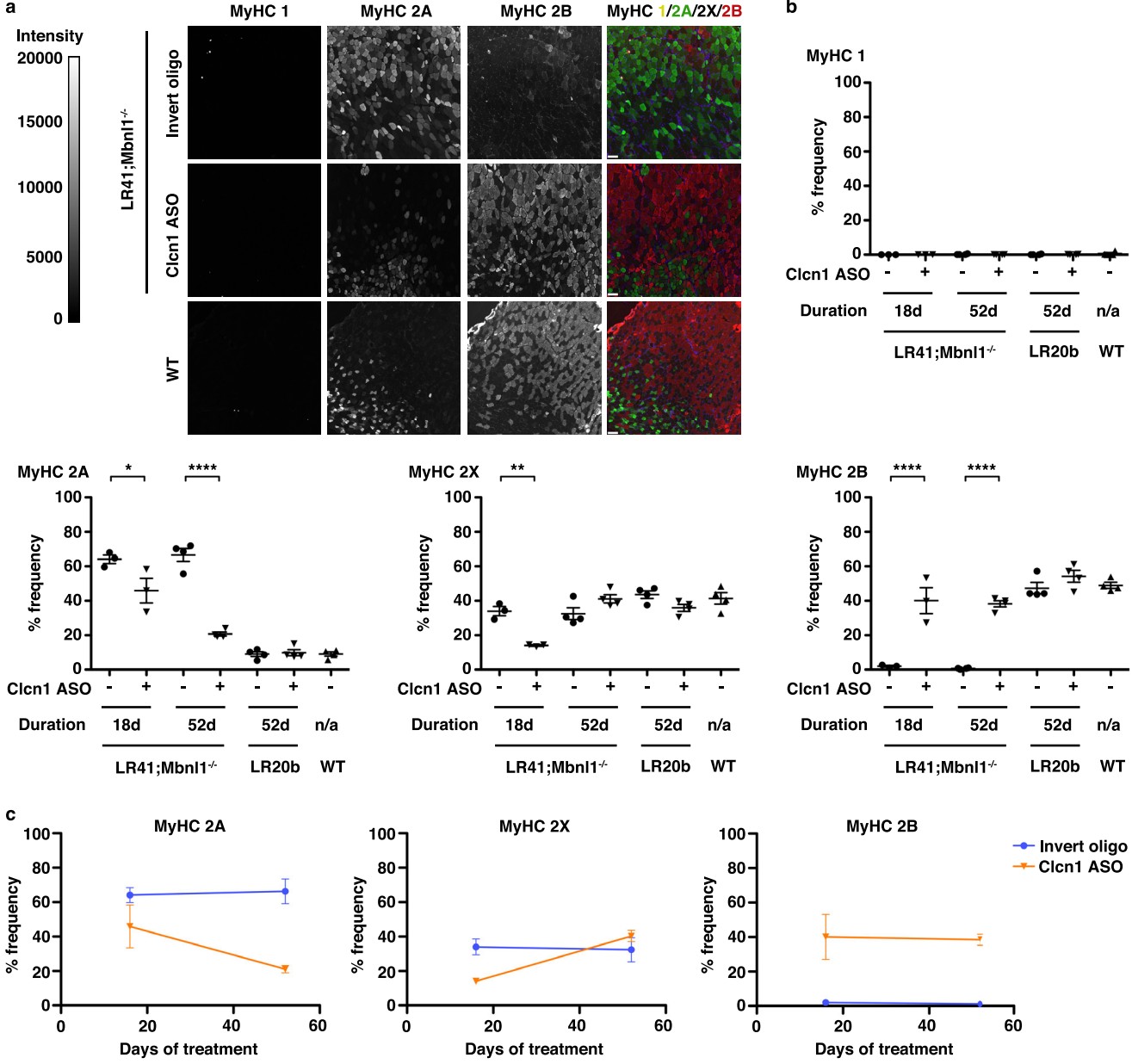

**Fig. 3 | Reversal of myosin fiber type transition in ASO-treated LR41;*Mbnl1*⁻/⁻ muscles.** We used immunofluorescence to quantify MyHC 1, MyHC 2 A, MyHC 2X, and MyHC 2B protein expression in tibialis anterior (TA) muscles of LR41;*Mbnl1*⁻/⁻ ($N = 7$) and LR20b ($N = 4$) treated with either the Clcn1 ASO (+) or the invert oligo (−). Untreated wild-type (WT; $N = 4$) TA muscles served as controls. **a** Representative images of LR41;*Mbnl1*⁻/⁻ treated with either the Clcn1 ASO or invert oligo for 18 days. Fluorescence intensity is 0–20,000 gray scale units. In the merge

images, MyHC 1 is colored yellow, MyHC 2 A green, MyHC 2X black (unlabeled), and MyHC 2B red. Size bars = 100 μm. **b** Quantification of MyHC 1, 2 A, 2X, and 2B fibers in TA muscles treated for 16–18 days or 52 days and untreated WT controls. ****$P < 0.0001$; **$P < 0.01$; *$P < 0.05$ (one-way ANOVA). **c** Time course of MyHC 2 A, 2X, and 2B expression in LR41;*Mbnl1*⁻/⁻ TA muscles treated with invert oligo (blue) or Clcn1 ASO (orange). Error bars indicate ±s.e.m. Source data are provided as a Source Data file.

proteins. In LR41;*Mbnl1*⁻/⁻ muscles treated with the control oligo, MyHC 2 A predominated at over 60% while 2B fibers were nearly absent (Fig. 3a–c). By contrast, in muscles treated with the Clcn1 ASO, MyHC 2 A was below 50% by 16–18 days of treatment, and about 20% after 52 days ($P < 0.05$ and $<0.0001$, respectively), while MyHC 2B was 40% at each time point ($P < 0.0001$). The frequency of MyHC 2X dropped from a baseline of around 35% to 15% by 16–18 days ($P < 0.01$), and had risen to 40% after 52 days. In LR20b muscles treated with Clcn1 ASO, the frequency of MyHC 2X was lower and MyHC 2B higher than in invert oligo-treated controls, although both were statistically non-significant, while the frequency of MyHC 2 A appeared similar to wild-type. MyHC 1 was low or absent in all experimental groups and wild-type controls.

ClC-1, the protein product of normally spliced *Clcn1* transcripts, was evident at the sarcolemma of several MyHC 2 A, 2X, and 2B fibers in Clcn1 ASO-treated muscles, but was absent in invert oligo-treated muscles (Fig. 4). In wild-type controls, the sarcolemmal signal for ClC-1 appeared more intense in MyHC type 2 A and 2X fibers than in 2B fibers. The cytoplasmic immunofluorescence signal also appeared higher in Clcn1 ASO-treated and wild-type muscles, as compared to invert oligo-treated muscles.

Following muscle injury, expression of embryonic myosin, *Myh3*, is upregulated transiently in newly regenerated fibers before being replaced by adult myosin isoforms. To determine the treatment effect on myosin transcript levels, we used ddPCR to quantify RNA expression of the genes *Myh2* (MyHC 2 A), *Myh1* (MyHC 2X), *Myh4* (MyHC 2B),

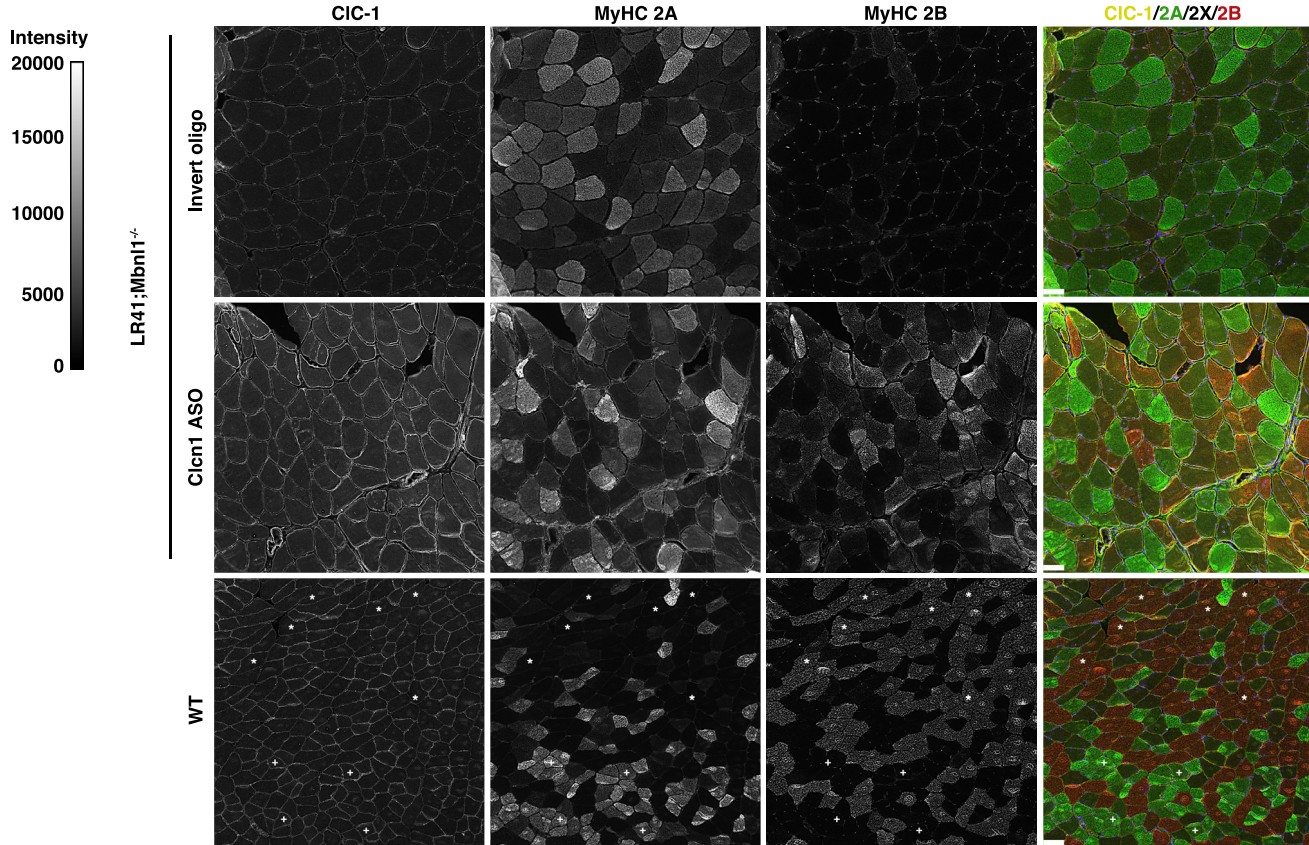

**Fig. 4 | Localization of ClC-1 protein.** We used antibodies targeting CLC-1, MyHC 2 A, and MyHC 2 B proteins to label TA muscles of LR41;*Mbnl1*[−/−] treated with Clcn1 ASO or invert oligo for 16 days and untreated WT controls (*N* = 4 each group). Representative images of each are shown. Fluorescence intensity is 0–20,000 gray scale units. In the merge images, ClC-1 is colored yellow, MyHC 2 A green, MyHC 2X black (unlabeled), and MyHC 2B red. In the WT images, "+" indicates 2 A fibers and "*" indicates 2X fibers in order to highlight the lower ClC-1 intensity evident at the membrane in 2B fibers vs. 2 A or 2X fibers. Size bars = 50 μm.

and the muscle injury marker *Myh3* (MyHC-emb). Using the copies/μl data for the four transcripts, we calculated the percent expression for each individual transcript. In wild-type TA muscles, *Myh2* was <1%, *Myh1* about 20%, and *Myh4* nearly 80% of total myosin gene expression (Fig. 5a, b). In LR41;*Mbnl1*[−/−], treatment with Clcn1 ASO reduced *Myh2* levels from nearly 20% down to <5% (*P* < 0.0001), reduced *Myh1* levels from nearly 80% down to about 50% (*P* < 0.0001 and <0.01), and increased *Myh4* 13-fold from <4% up to 45% (*P* < 0.0001), as compared to invert oligo-treated muscles. In LR20b, treatment with Clcn1 ASO reduced *Mhy1* from 36% to 16% (*P* < 0.05) and increased *Myh4* from 63% to 83% (*P* < 0.05) vs. invert control oligo-treated muscles, while *Myh2* remained <1% regardless of treatment. *Clcn1* exon 7a inclusion strongly correlates with myosin gene expression by ddPCR (r = 0.80, 0.91, and −0.93; *P* < 0.0001; Fig. 5c). In muscles treated with the Clcn1-targeting ASO, expression of the muscle injury marker *Myh3* was 70–80% lower than in control-treated muscles (*P* < 0.01) and also correlated with *Clcn1* alternative splicing (r = 0.74; *P* < 0.0001) (Fig. 5d–f).

### MyHC 2 A/2X hybrid fibers in LR41;*Mbnl1*[−/−]
Next we examined the relationship between myosin transcripts and the corresponding proteins. The frequency of MyHC 2 A by immuno-fluorescence was consistently higher than *Myh2* gene expression by ddPCR, although they strongly correlated (r = 0.90; *P* < 0.0001) (Fig. 6a). The combined frequencies of *Myh2* and *Myh1* strongly correlated with the combined frequencies of MyHC 2 A and 2X (r = 0.93; *P* < 0.0001), as did *Myh4* transcripts and MyHC 2B protein (r = 0.93; *P* < 0.0001).

We observed variable intensity of immunolabeling for MyHC 2 A in LR41;*Mbnl1*[−/−] TA muscles, similar to that in the gastro-cnemius muscles. Hybrid fibers contain a mixed composition of two or more MyHC isoforms and are frequent in muscles under-going fiber type transformation[15]. Prior studies also have shown an increased proportion of individual fibers expressing more than one myosin isoform in human DM muscle[34–36]. To explore the possibility that the variable labeling intensity of MyHC 2 A and/or the mismatch of *Myh2* RNA with MyHC 2 A protein may indicate the presence of hybrid fibers, we co-labeled TA muscle tissue sections with antibodies targeting both MyHC 2 A and 2X. In LR41;*Mbnl1*[−/−], more than half of the fibers expressing MyHC 2 A also express MyHC 2X, regardless of treatment (Fig. 6b–d). Pure 2 A and 2 A/2X hybrid fibers were reduced ~70% in muscles that received the Clcn1 ASO, as compared to those that received the invert oligo. In WT TA muscles examined, about half of the total 2 A fibers were 2 A/2X hybrids.

### Clcn1 ASO treatment reduces hypertrophy in LR41;Mbnl1[−/−]
In adult humans with DM1, oxidative and glycolytic muscle fiber size increases from the 2nd through 5th decade[21]. To determine the effect of severe myotonia on myofiber size in LR41;*Mbnl1*[−/−] mice, we measured minimal Feret's diameter[37] in TA muscle tissue sections. In muscles treated with the invert control oligo, the size distribution of all fiber types was shifted toward hypertrophy, with mean diameter of MyHC 2 A fibers was 75% greater (*P* = .0002), MyHC 2X 40% greater (*P* = 0.004), and MyHC 2B 20% greater (non-significant) than in wild-type control muscles (Fig. 7a, b). Treatment with Clcn1 ASO shifted the size

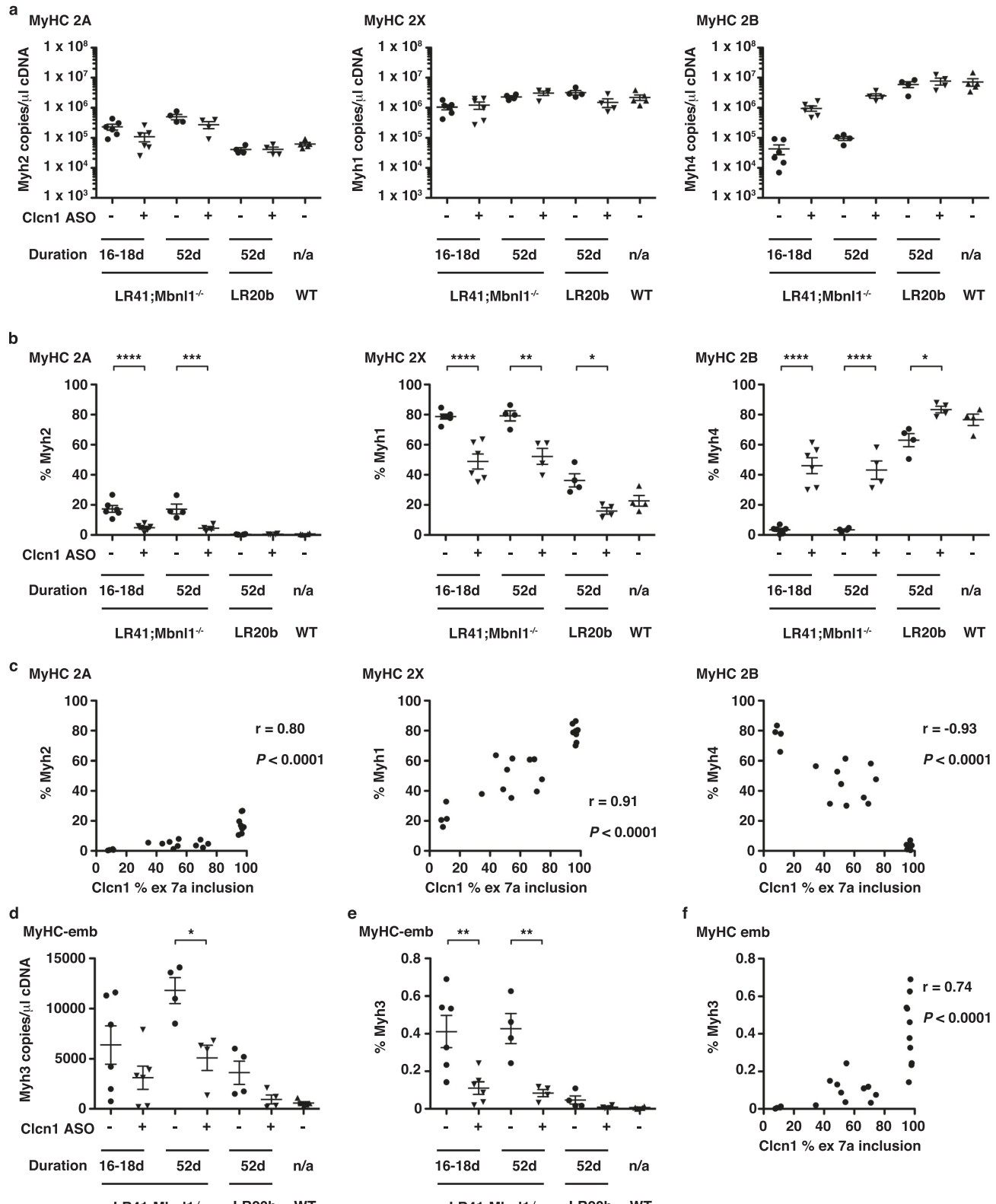

distribution and reduced mean diameter of MyHC 2 A fibers by ~12% and of 2X fibers by 16% as compared to muscles treated with invert control oligo, and rescued 2B fiber diameter to wild-type values ($P = 0.028$).

## MyHC fiber type mediates *ACTA1*-CUG^exp transcript abundance

In prior studies, alternative splicing of *Apt2a1* served as a sensitive indicator of *ACTA1*-CUG^exp transcript levels and therapeutic drug

activity[30,38]. Having shown that selective correction of *Clcn1* splicing exacerbated deregulated splicing of *Atp2a1* in LR20b mice, we then examined the effect of *Clcn1* splicing correction on expression of *ACTA1*-CUG^exp transcripts in treated muscles. Using ddPCR, we found that *ACTA1*-CUG^exp mRNA transcript levels doubled after 16–18 days and were increased 65% after 52 days of treatment with *Clcn1* ASO, as compared to invert oligo-treated controls ($P < 0.01$ and $< 0.05$,

**Fig. 5 | ddPCR quantification of MyHC gene expression in treated muscles.**
**a** Using ddPCR, we quantified gene expression of *Myh2* (left; MyHC type 2 A), *Myh1* (middle; MyHC Type X), and *Myh4* (MyHC Type 2B) as copies/μl cDNA in TA muscles of LR41;*Mbnl1*[-/-] (*N* = 10) and LR20b (*N* = 4) mice treated with Clcn1 ASO (+) or invert oligo (−) for 16, 18, or 52 days. Untreated wild-type (WT; *N* = 5) TA muscles served as controls. **b** Using the copies/μl data in **a**, we calculated the percentage (%) of total myosin transcripts for *Myh2* (left), *Myh1* (middle), and *Myh4* (right) of LR41;*Mbnl1*[-/-] (*N* = 10), LR20b (*N* = 4), and WT (*N* = 4). **c** Relationship of ddPCR quantification of *Clcn1* exon 7a inclusion (x-axis; see Fig. 2) with percentage of *Myh2*, *Myh1*, and *Myh4* (y-axis) in individual samples of LR41;*Mbnl1*[-/-] (*N* = 10) and WT (*N* = 4). The Pearson

correlation coefficient r and *P*-value for each are shown. **d** Quantification of embryonic myosin *Myh3* (MyHC-emb) gene expression, a marker of muscle regeneration, in LR41;*Mbnl1*[-/-] (*N* = 10), LR20b (*N* = 4), and WT (*N* = 5). **e** *Myh3* expression as percentage of total myosin gene expression in LR41;*Mbnl1*[-/-] (*N* = 10), LR20b (*N* = 4), and WT (*N* = 5). **f** Relationship of ddPCR quantification of *Clcn1* exon 7a inclusion (x-axis) with *Myh3* as percentage of total myosin (y-axis) in individual samples of LR41;*Mbnl1*[-/-] (*N* = 10) and WT (*N* = 4). The Pearson correlation coefficient r and *P*-value for each are shown. ****$P < 0.0001$; ***$P < 0.001$; **$P < 0.01$; *$P < 0.05$ (one-way ANOVA). Error bars indicate ±s.e.m. Source data are provided as a Source Data file.

respectively) (Fig. 8a, b). To explore whether this elevated mRNA abundance could be due to increased transcription of the transgene, we measured *ACTA1*-CUG[exp] pre-mRNA, and found that it also was elevated, with increases of 55% and 70% after 16–18 and 52 days, respectively ($P < 0.05$ and <0.01). By contrast, expression of *Dmpk*, the gene that carries the CUG[exp] repeat expansion in human DM1, appeared similar in muscles regardless of treatment (Fig. 8c).

To determine the relationship between expression of the *ACTA1* transgene and myosin isoforms, we compared *ACTA1* mRNA and pre-mRNA with *Myh2*, *Myh1*, and *Myh4* transcript abundance. *ACTA1* mRNA and pre-mRNA inversely correlated with *Myh2* (r = −0.70 mRNA and −0.65 pre-mRNA; $P = 0.038$ and 0.078) and *Myh1* (r = −0.85 and −0.81; $P = 0.007$ and 0.014) frequency, and correlated with *Myh4* (r = 0.89 and 0.83; $P = 0.003$ and 0.011) frequency at both the early and later time points (Fig. 8d, e).

## Discussion

In DM1, myotonia and oxidative fibers tend to be most prominent in muscles that are the weakest, suggesting that repetitive action potentials may contribute to muscle pathology. In this study we use three mouse models of DM1 to establish that chronic myotonia causes muscle fiber type transition from glycolytic to oxidative, that the transition is reversible by correction of the underlying molecular defect, and that alternative splicing of *Clcn1* predicts fiber type proportion. The reversibility of fiber type transitions in DM1 mice argues against selective loss of glycolytic fibers as the cause of oxidative fiber predominance observed in adult DM muscle biopsies[20,21]. The improvement in fiber diameter and reduction of muscle injury in ASO-treated mice indicate a therapeutic benefit of myotonia reduction and supports further development of *Clcn1*-targeted therapies for DM1. By elimination of the premature termination codon, a Clcn1 exon skipping ASO increases the overall abundance of ClC-1 protein, functional ClC-1 channels, and chloride conductance[24]. Potential clinical benefits of specific anti-myotonia treatment in DM1 include, (1) decreased muscle relaxation time, (2) restored MyHC fiber type pattern, (3) increased muscle power due to a higher proportion of mechanically stronger glycolytic fibers, (4) reduced muscle injury, (5) improved fiber size, and (6) enhanced overall muscle function.

ASO and small molecule therapies that target the pathogenic *DMPK*-CUG[exp] transcripts are currently in development for DM1. A clear advantage of this approach is the potential to treat most or all of the clinical features of DM1, including myotonia, with a single drug. A potential hurdle relates to the instability of the CTG repeat expansion in somatic tissues, reaching around 3000–6000 repeats, corresponding to around 9000–18,000 nucleotides[39–41]. Accordingly, candidate antisense chemistries that target the CUG[exp] sequence using a MBNL protein displacement strategy[42] may require several dozen or more copies of a 25-residue oligo to bind each *DMPK*-CUG[exp] transcript in order to release enough MBNL protein to have a therapeutic effect, creating a potential for dose limiting toxicity and suboptimal clinical benefit. Antisense strategies that target the CUG[exp] sequence for cleavage and subsequent degradation carry the risk of inadvertent silencing of other genes that also contain CUG repeats, while those strategies that target sequence outside of

the repeat compromise selectivity for the expanded repeat-containing transcripts, potentially silencing the non-expanded *DMPK* allele and risk exacerbating clinical features that may arise from haploinsufficiency. In addition, an early phase clinical trial of an RNase H-active ASO for the treatment of myotubular myopathy recently was terminated due to poor tolerability at the low dose (clinicaltrials.gov identifier NCT04033159), suggesting that translation of a cleavage-based approach to muscle may be difficult. By contrast, the splice-shifting morpholino chemistry that we have used in this study has demonstrated safety in humans, with four drugs already FDA-approved for the treatment of Duchenne muscular dystrophy[43]. Several molecular conjugates have shown promise to facilitate the delivery and potency of multiple ASO chemistries in skeletal muscle tissue[30,44–50]. To enhance therapeutic benefit in DM1, drugs directed toward *DMPK*-CUG[exp] transcripts could be combined with an adjuvant *Clcn1*-targeting ASO.

A recent clinical trial showed that treatment with the sodium channel blocking drug mexiletine for six months reduced hand grip myotonia in DM1 subjects vs. a placebo-treated group[51]. The same study also found a possible improvement in 6 min walk test in the mexiletine group after three months, when the drug was detectable in the serum of 90% of subjects, and a non-significant trend for improvement at 6 months, when mexiletine was detectable in the serum of only 65% of subjects, suggesting that non-compliance with dosing may have obscured a potential benefit at the later time point. The relatively short half-life of mexiletine generally requires dosing three or four times per day, which may limit long-term compliance. It's unclear whether a sufficiently high dose of mexiletine could be achieved and maintained to enable a continuous and lasting reduction of myotonia necessary to reverse fiber type transition. Another disadvantage of mexiletine is that it has pro-arrhythmic effects on the heart, which complicates it use in individuals with DM1 who already are prone to cardiac conduction disturbance[52].

The greater abundance of *ACTA1*-CUG[exp] transcripts that we observed in *Clcn1* ASO-treated muscles could be interpreted as cause for concern in the translation of this therapeutic approach to DM1 patients, perhaps by increasing the production and/or enhancing the stability of *DMPK*-CUG[exp] RNA in skeletal muscle. This is unlikely for several reasons, (1) the elevation of *ACTA1*-CUG[exp] pre-mRNA and mRNA to a similar degree in ASO-treated muscles suggests that increased transcription of the *ACTA1* transgene is the mechanism, (2) the strong negative correlation of *ACTA1*-CUG[exp] transcript levels with *Myh2* (2 A) and *Myh1* (2X) transcripts, and strong positive correlation with *Myh4* (2B) suggest that expression of the human *ACTA1* transgene is higher in 2B fibers and that the increase of *ACTA1*-CUG[exp] RNA observed in ASO-treated muscles was an artifact of the human *ACTA1* transgene responding to the fiber type shift, and (3) transcript levels of *Dmpk*, the gene that carries the expanded repeat in human DM1, appeared unchanged in treated muscles, arguing against a similar *Clcn1*-targeting ASO leading to increased *DMPK*-CUG[exp] expression. In fact, the therapeutic effects that we observed in the setting of higher *ACTA1*-CUG[exp] RNA content suggests the potential for an even greater clinical benefit of fiber type transition in DM1 patients if *DMPK*-CUG[exp] expression remains stable.

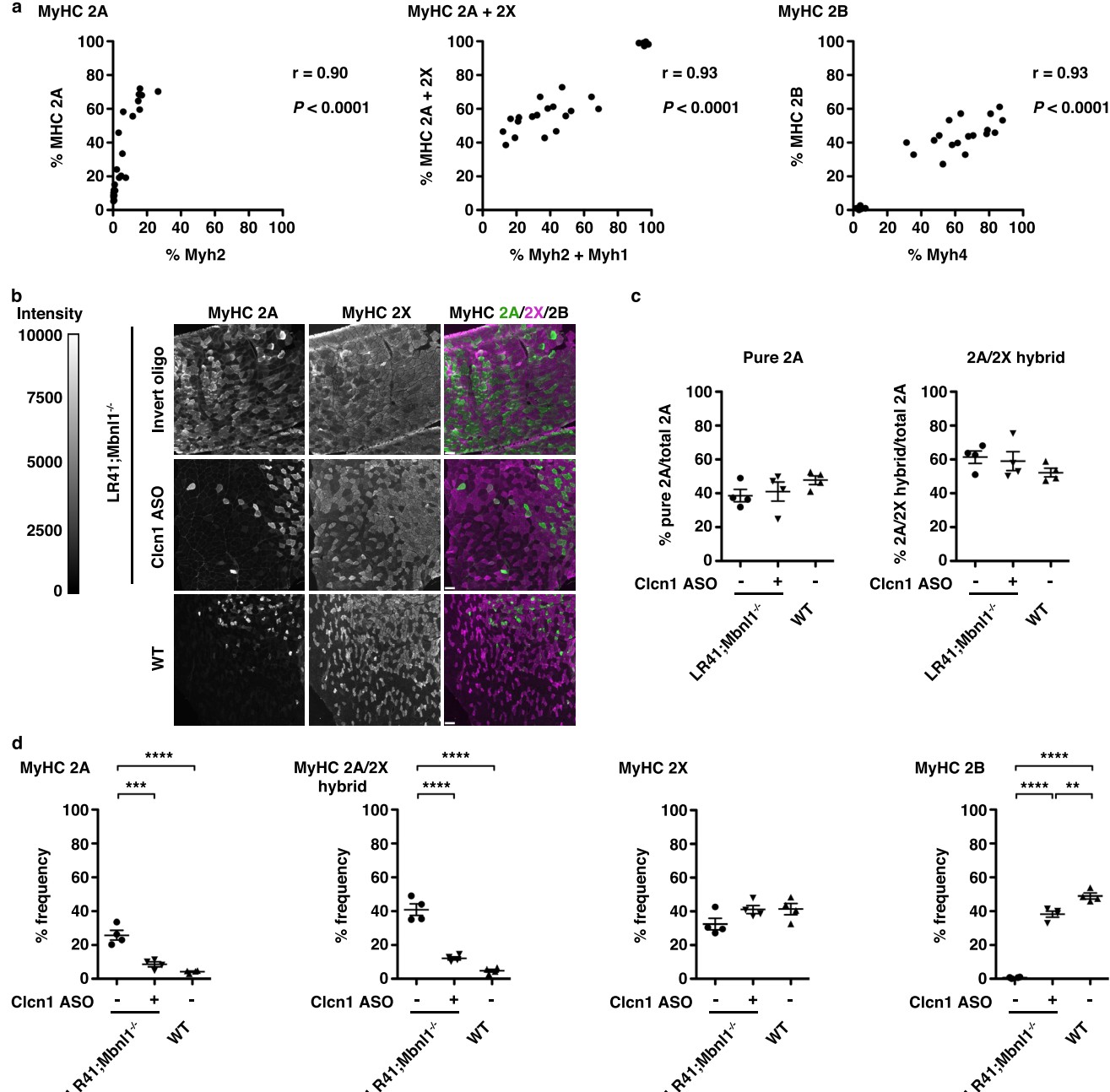

**Fig. 6 | Quantification of MyHC 2 A/2X hybrid fibers. a** Relationship of *Myh2* (left), *Myh2* combined with *Myh1* (middle), and *Myh4* (right) transcript expression by ddPCR (*x*-axes) with quantification of the corresponding proteins MyHC 2 A (left), MyHC 2 A combined with MyHC 2X (middle), and MyHC 2B (right) by immunofluorescence (IF) (*y*-axes) in LR41;*Mbnl1*⁻/⁻ (*N* = 7), LR20b (*N* = 4), and WT (*N* = 3). The Pearson correlation coefficient r and *P*-value for each are shown. **b** We used antibodies targeting MyHC 2 A and 2X and immunofluorescence (IF) analysis of MyHC protein expression in TA muscles of LR41;*Mbnl1*⁻/⁻ treated with either the Clcn1 ASO or the invert oligo (*N* = each group). Untreated wild-type (WT) TA muscles (*N* = 4) served as controls. Representative images of each are shown. Fluorescence intensity is 0–10,000 gray scale units. In the merge images, MyHC 2 A appears green, MyHC 2X magenta (pseudo-colored), and MyHC 2B black (unlabeled). Fibers that show signal for both MyHC 2 A and MyHC 2X are MyHC 2 A/2X hybrid fibers. Size bars = 100 μm. **c** Quantification (%) of overall fibers expressing 2 A without 2X (pure 2 A; left) and 2 A that co-express 2X (2 A/2X hybrids) in LR41;*Mbnl1*⁻/⁻ treated with either the Clcn1 ASO (+) or the invert oligo (−) and untreated WT controls (−) (*N* = 4 each group). **d** Quantification of pure MyHC 2 A (left), 2 A/2X hybrid (left-center), 2X (center-right), and 2B (right) as % frequency (*N* = 4 each group). ****$P < 0.0001$; ***$P < 0.001$; **$P < 0.01$ (one-way ANOVA). Error bars indicate ±s.e.m. Source data are provided as a Source Data file.

The nuclear co-localization of CUGᵉˣᵖ RNA with MBNL2 protein that we observed in LR41;*Mbnl1*⁻/⁻ suggests that the exacerbated phenotype in this model may result in part from compound MBNL loss of function, similar to the combined genetic reduction of *Mbnl1* and *Mbnl2* in mice[26,53]. The reduced intensity of nuclear inclusions in LR41;*Mbnl1*⁻/⁻ supports a previous report that MBNL1 protein is important for the formation of ribonuclear foci in DM1 cells[54]. In addition, the apparent absence of CUGᵉˣᵖ inclusions in several myonuclei also suggests the possibility that the exacerbated phenotype also may result in part from enhanced toxicity of CUGᵉˣᵖ transcripts, either through the binding of other nuclear proteins and/or localization to the cytoplasm. The increased abundance of MBNL2 protein that we found in myonuclei and interstitial cells of LR41;*Mbnl1*⁻/⁻ muscle tissue is consistent with prior observations of a

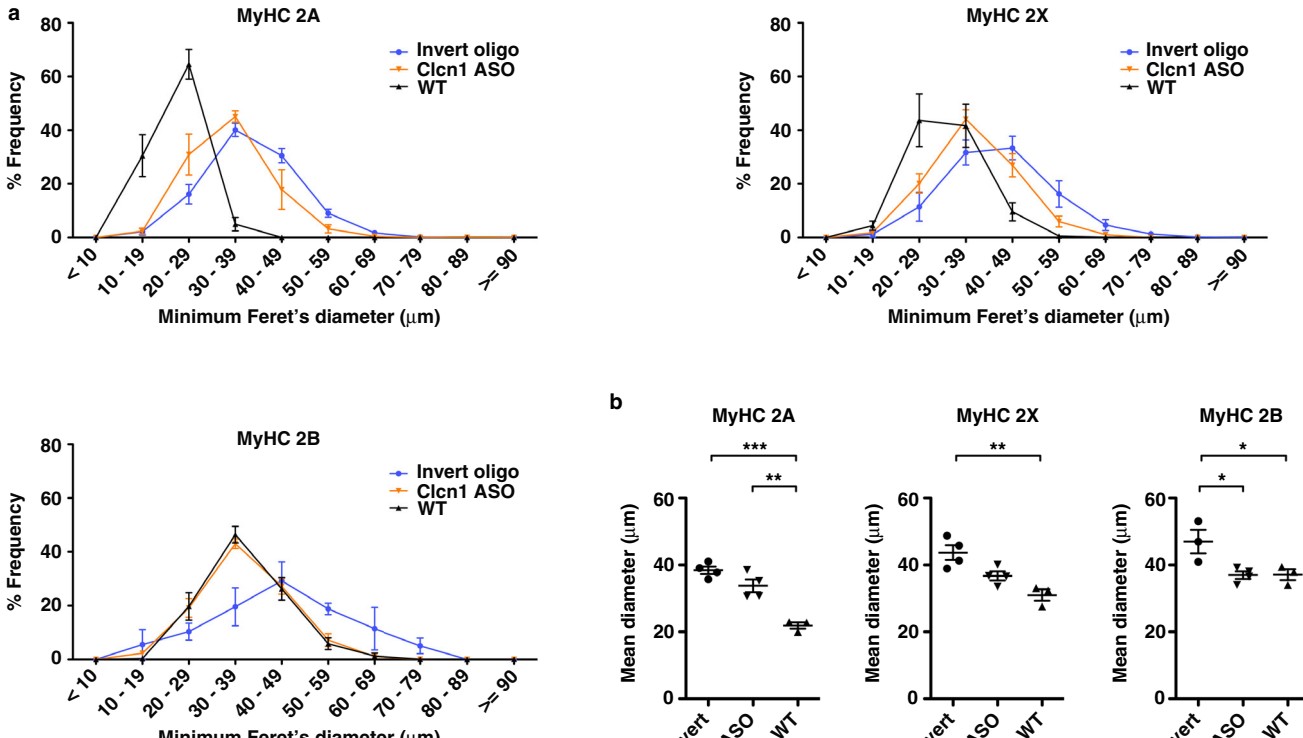

**Fig. 7 | Fiber diameter in treated muscles. a** We measured minimal Feret's diameter, defined as the minimum distance between parallel tangents[37], of tibialis anterior (TA) muscle fibers in LR41;*Mbnl1*⁻/⁻ treated with Clcn1 ASO or invert oligo (*N* = 4 each). Untreated wild-type (WT; *N* = 3) served as controls. Diameter is displayed as % frequency for MyHC 2 A (upper left), MyHC 2X (upper right), and MyHC 2B (lower left) fibers. **b** Mean fiber diameter using data from a). ***$P$ < 0.001; **$P$ < 0.01; $P$ < 0.05 (one-way ANOVA). Error bars indicate ±s.e.m. Source data are provided as a Source Data file.

compensatory increase of *Mbnl2* mRNA and protein expression in *Mbnl1*⁻/⁻ muscle[26].

The localization of ClC-1 protein within the muscle fiber is controversial, with evidence supporting that the majority of channels are located within the transverse tubular system of the cytoplasm[55], exclusively at the muscle membrane[56], or can change from cytoplasm to the membrane with age[57]. An incidental finding in our study was that ClC-1 protein expression appeared increased in both the cytoplasm and cell membrane in LR41;*Mbnl1*⁻/⁻ ASO-treated muscles, as compared to invert oligo-treated muscles. In wild-type muscles, we also observed that ClC-1 labeling at the sarcolemma appeared lowest in 2B fibers, which are reported to have higher ClC-1 protein expression and chloride conductance than oxidative fibers[15].

## Methods

### Experimental mice

The Massachusetts General Hospital (MGH) IACUC approved all mouse studies. The Human Skeletal Actin Long Repeat (HSA^LR) line 20b (LR20b) and line 41 (LR41) transgenic mouse model[22], and the muscleblind-like 1 knockout (*Mbnl1*⁻/⁻)[23] mouse model containing a homozygous deletion of *Mbnl1* exon 3 have been described previously. We obtained LR20b and LR41 transgenic mice from Dr. C. Thornton (University of Rochester) and *Mbnl1*⁻/⁻ mice from Dr. M. Swanson (University of Florida) via material transfer agreements. All mice were maintained on the FVB genetic background. Wild-type FVB mice (The Jackson Laboratory, strain number 001800) served as controls. Age ranged from 7 weeks to 18 months. All mice were fed standard chow (irradiated Prolab Isopro RMH 3000) ad libitum and housed in groups of up to five. Lighting was time controlled on a standard 12:12 light:dark cycle. Temperature and humidity were stable and consistent at 20–23 °C and 30–70%, respectively. Euthanasia was performed by 5% isoflurane anesthetic overdose with secondary cervical dislocation.

### Mouse genotyping

We crossed LR41 with *Mbnl1* heterozygotes to generate mice that are LR41 homozygous and Mbnl1 heterozygotes (LR41;*Mbnl1*⁺/⁻), and then crossed LR41;*Mbnl1*⁺/⁻ mice to generate double homozygous LR41;*Mbnl1*⁻/⁻ mice. At weaning we performed tail biopsy and isolated genomic DNA from biopsy specimens using a commercially available tissue/blood DNA isolation kit (Qiagen product number 69506). To determine zygosity of the *ACTA1* transgene, we used a PCR assay with two sets of primers, one specific for mouse *Acta1* and the other specific for human *ACTA1* in the same reaction, and quantitative band densitometry to calculate the *ACTA1*/*Acta1* ratio. To detect the *Mbnl1* exon 3 deletion allele, we modified a previously published protocol[23] that uses a common left primer and separate right primers in separate PCR reactions, one to detect the *Mbnl1* exon 3 deletion and the other to detect the wild-type allele. Primer sequences are shown in Supplementary Table 1.

### Estimation of the *ACTA1* transgene CTG repeat length

Using a previously published protocol[25], we digested 100 ng genomic DNA with the restriction enzyme HindIII-HF (New England Biolabs product number 3104), used 10 ng of digested DNA as template for PCR with primers spanning the CTG repeat insertion, separated PCR products by 1.5% agarose gel electrophoresis, stained gels using 1× SYBR I green nucleic acid gel stain (Life Technologies product number S7567), and estimated CTG repeat length by subtracting the number of nucleotides amplified by the primers upstream and downstream of the CTG repeat from the total amplicon size, and dividing by three.

### Activity monitoring assay

We modified our previously published protocol[25], as follows. All mice were tested at night so that at the time of testing, they were behaviorally most active, and were acclimatized to the test room for at least an

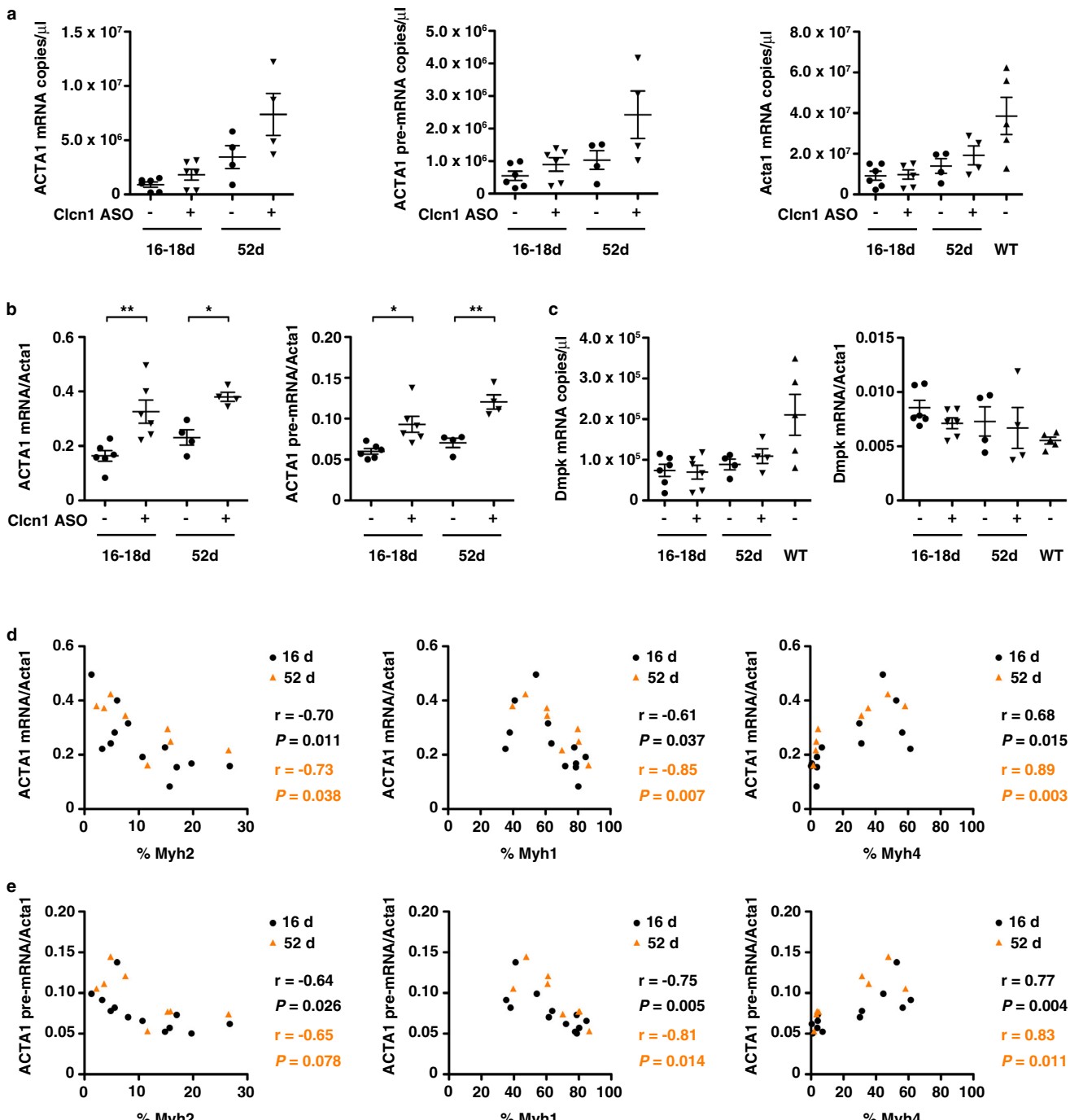

**Fig. 8 | Relationship of *ACTA1*-CUG^exp transgene and myosin gene expression in treated muscles. a** We used ddPCR to quantify expression of human *ACTA1*-CUG^exp mRNA, human *ACTA1*-CUG^exp pre-mRNA, and reference gene mouse *Acta1* in TA muscles of LR41;*Mbnl1*^−/− (*N* = 10) treated with Clcn1 ASO (+) or invert oligo (−) as copies/μl cDNA. Untreated WT TA muscles (−) (*N* = 4) served as controls. **b** Using the copies/μl cDNA data in **a**), we normalized expression of *ACTA1*-CUG^exp mRNA (left) and *ACTA1*-CUG^exp pre-mRNA (right) to mouse *Acta1* in LR41;*Mbnl1*^−/− (*N* = 10). **P < 0.01; *P < 0.05 (one-way ANOVA). **c** ddPCR quantification of *Dmpk* expression

in treated muscles as copies/μl cDNA (left) and normalized to *Acta1* (right) n LR41;*Mbnl1*^−/− (*N* = 10) and WT (*N* = 4). **d** Relationship of % *Myh2* (left), *Myh1* (middle), and *Myh4* (right) gene expression (*x*-axes) with *ACTA1*-CUG^exp mRNA normalized to *Acta1* and **e** with *ACTA1*-CUG^exp pre-mRNA (*y*-axes) in LR41;*Mbnl1*^−/− TA muscles (*N* = 10) treated for 16−18 days (black circles) or 52 days (orange triangles). The Pearson correlation coefficient r and *P*-value for each are shown. Error bars indicate ±s.e.m. Source data are provided as a Source Data file.

hour. We monitored spontaneous activity using individual chambers in sets of two, for 30 ×1 min intervals in an isolated room at the same time every day, and with the same handler (Opto-Varimex 4 Activity Meter; Columbus Instruments, Inc). This system uses a grid of invisible infrared light beams that traverse the animal chamber front to back and left to right to monitor the position and movement of the mouse

within an X-Y-Z plane. The sensor pairs consist of an infrared emitter bar with eight infrared beams and a matching detector bar. Every interruption of one of the optimal beams is registered as a count. Vertical counts are the total counts from the *z*-axis sensors. Vertical breaks are single rearing events with one second between each event. Ambulatory time is the quantification of the total time in seconds (sec)

spent making large ambulatory movement. Rest time is defined as the combined period of time (sec) without detection of movement. Stereotypic time (sec) is the quantification of total time spent on small rapid movements such as scratching, grooming, or other stereotypic non-ambulatory movements. The total distance traveled during the interval is recorded in centimeters (cm). For each parameter, we used average values of three separate trials on three separate days for comparison between groups.

## Chronic moderate intensity exercise training

We exercised 14–14.5-month-old HSA[LR] mice using an adjustable variable speed belt treadmill (Exer 3/6; Columbus Instruments, Inc.) on a flat surface (no incline) that began with acclimatization at 3 m/min for 2 min, acceleration from 3 m/min to 11.5 m/min over 1 min, followed by 11.5 m/min 30 min, and a 1 min cool down period of the speed gradually slowing to a stop, for a total of 34 min regimen on 6 days per week for 15 weeks, as previously published[25]. A 15-day acclimatization began with a speed of 11.5 m/min for 3 min and increased duration by 2 min per day until reaching the target duration of 30 min per session. HSA[LR] mice started the acclimatization procedure at age 14 months and completed the training regimen at age 17.5–18 months.

## Fluorescence in situ hybridization

To identify CUG[exp] RNA foci, we used our modification of a previously published protocol[25,58]. We fixed 8 μm muscle cryosections with 3% paraformaldehyde, pH 7.3 in 1× PBS, washed in 1× PBS, permeabilized nuclei using 0.5% Triton X-100 in 1× PBS for 5 min or 0.2% Triton X-100 in 1× PBS for 10 min at room temperature, and incubated in pre-hybridization solution (30% formamide/2X SSC) for 10 min at room temperature. Hybridization was for 3 h at 37 °C in 33% formamide, 2× SSC, 0.2 mg/ml bovine serum albumin (NEB product number B9001S), 70 μg/ml yeast t-RNA (Invitrogen product number 15401-011), 2 mM ribonucleoside vanadyl complex (NEB product number S1402S), and 1 μg/ml 2′-O-methyl-modified RNA CAG repeat probe 5′ labeled with Alexa647. Post-hybridization, we incubated sections in 30% formamide/2X SSC for 30 min at 42 °C followed by 1× SSC for 30 min at room temperature, and three washes in 1X PBS at room temperature before proceeding to labeling with primary antibody.

**Hybridization probe.** 5′-Alexa647-mCmAmGmCmAmGmCmAmGmCmAmGmCmAmGmCmAmGmCmA-3′

(HPLC-purified; "m" designates that RNA bases have 2′-O-methyl modifications; IDT).

## Immunolabeling and muscle fiber morphometry

To localize MBNL1 and MBNL2 proteins after FISH, we incubated muscle sections in anti-MBNL1 rabbit polyclonal antibody (2 μg/ml in PBS; Abcam product number ab45899) together with anti-MBNL2 (3B4) mouse monoclonal antibody (isotype IgG2b; 4 μg/ml PBS; Santa Cruz Biotechnology, Inc. product number sc-136167) overnight at 4 °C followed by goat anti-rabbit Alexa 488 (Invitrogen product number A-11034) and goat anti-mouse IgG2b Alexa 546 (Invitrogen product number A-21143) secondary antibodies (1 μg/ml PBS each) for 1 h at room temperature. Post-FISH, some sections were labeled with only the anti-MBNL2 primary and secondary antibodies. To highlight muscle fibers and nuclei, we added FITC-labeled wheat germ agglutinin (10 μg/ml; Sigma product number L4895) and DAPI (33 ng/ml) together with the secondary antibody.

For determination of myosin fiber types, we blocked unfixed 8 μm frozen muscle sections with 10% normal goat serum in PBS for one at room temperature followed by incubation in mouse monoclonal primary antibodies BA-F8 (isotype IgG2b; myosin heavy chain Type 1), SC-71 (isotype IgG1; myosin heavy chain Type 2 A), and BF-F3 (isotype IgM; myosin heavy chain Type 2B), each at 5 μg/ml PBS for 1 h at room temperature (Developmental Studies Hybridoma Bank, University of Iowa; all deposited by S. Schiaffino, University of Padova). Fibers expressing myosin heavy chain Type 2X remained unlabeled (black) or showed <50% intensity of the 2 A or 2B fibers. We also identified 2× fibers using mouse monoclonal antibody 6H1 (isotype IgM) at 5 μg/ml PBS (Developmental Studies Hybridoma Bank, University of Iowa; deposited by C. Lucas, University of Sydney). Secondary antibodies were Alexa647 goat anti-mouse IgG2b (Invitrogen product number A-21242), Alexa 488 goat anti-mouse IgG1 (Invitrogen product number A-21121), and Alexa 546 goat anti-mouse IgM (Invitrogen product number A-21045), all at 2 μg/ml PBS for 1 h at room temperature. To localize ClC-1 protein, we blocked unfixed 8 μm frozen sections with 10% normal goat serum in PBS and labeled with an affinity purified anti-rat CLC-1 IgG rabbit polyclonal antibody (20 μg/ml in PBS; Alpha Diagnostic International product number CLC11-A) together with anti-MyHC 2 A (SC-71) and anti-MyHC 2B (BF-F3) antibodies (see above) overnight at 4 °C. Secondary antibody was goat anti-rabbit Alexa647 (1 μg/ml; Invitrogen product number A-21244). We mounted all slides using anti-fade medium (Prolong Gold; Invitrogen product number P36930) and No. 1 1/2 cover glasses (Zeiss product number 474030-9000-000), as previously described[30].

## Quantitative fluorescence microscopy

To capture single images or z-series stacks, we used an AxioImager microscope (Zeiss), filters for DAPI (excitation/emission 365/445; Zeiss filter set 49), GFP (excitation/emission 470/525; Zeiss filter set 38), Cy3 (excitation/emission 550/605; Zeiss filter set 43 HE), and Cy5 (excitation/emission 640/690; Zeiss filter set 50), a Flash 4.0 LT sCMOS camera (Hamamatsu), and Volocity image acquisition software (version 6.3.1; Perkin Elmer). Objectives were ×5 EC Plan-NEOFLUAR NA 0.16, ×10 EC Plan-NEOFLUAR NA 0.3, ×20 Plan-APOCHROMAT NA 0.8, ×40 Plan-APOCHROMAT NA 1.4, and ×63 Plan-APOCHROMAT NA 1.4. To quantitate fluorescence, we used Volocity quantitation and restoration software modules (version 6.3.1; Perkin Elmer), as previously described[30]. We measured minimal Feret's diameter, defined as the minimum distance between parallel tangents[37], by MyHC fiber type in 8 μm frozen sections of TA muscle fibers of LR41;*MbnlI*[−/−] mice treated with the Clcn1 ASO or invert oligo. Untreated WT served as controls.

## Systemic antisense oligonucleotide (ASO) treatment

We used ASO 445236, a 20-mer RNase H-active gamper that targets *ACTA1* transcripts 3′ of the expanded CUG repeat region[38]. The central gap segment consists of ten 2′-deoxyribonucleotides (underlined below) that are flanked on the 5′ and 3′ wings by five 2′-*O*-methoxyethyl-modified nucleotides. The internucleotide linkages are phosphorothioate, and all cytosine residues are 5′-methylcytosines. We treated mice with a dose of 25 mg/kg by subcutaneous injection twice weekly for 4 weeks (eight doses) and again once every four weeks (two additional doses for a total of ten doses).

Gamper ASO 445236 sequence: 5′-CCATTTTCTTCCACAGGGCT-3′ (published previously[38]).

## Intramuscular ASO treatment

To correct *Clcn1* alternative splicing, we administered a morpholino ASO designed to suppress inclusion of Clcn1 exon 7a by intramuscular injection of tibialis anterior (TA) muscles, as described previously[24]. The contralateral TA received the 5′-to-3′ invert of the active antisense drug. We purchased both the Clcn1-targeting ASO and the invert control oligo from Gene Tools, LLC. The dose administered was 20 μg of each. To enhance distribution of antisense morpholinos, we pre-treated TA muscles by injecting with 0.4 U/L hyaluronidase 2 h prior to oligo injection. To facilitate intracellular uptake of oligos into the target tissue, we electroporated muscles immediately after IM injection using settings of 100 V/cm, pulse duration of 20 milliseconds, a frequency of 10 Hz, and total of 10 pulses per muscle.

Clcn1-7a antisense sequence: 5′-CCAGGCACGGTctgcaacagagaag-3′ (upper case denotes exon-targeting sequence, lower case intron-targeting sequence)

Clcn1-7a invert sequence: 5′-gaagagacaacgtctggcacggacc-3′ (non-targeting control)

Both of these oligos have been published previously[24].

## RNA isolation

We homogenized muscle tissues in Trizol (Life Technologies), removed DNA and protein using bromochloropropane, precipitated RNA with isopropanol, washed pellets in 75% ethanol, and dissolved pellets in molecular grade water according to manufacturer recommendations. To determine RNA concentration and quality, we measured A260 and A280 values (Nanodrop) and examined 18 S and 28 S ribosomal RNA bands by agarose gel electrophoresis, as described previously[30].

## RT-PCR analysis of alternative splicing

We made cDNA using Superscript III reverse transcriptase (Life Technologies) and random primers, and performed PCR using Amplitaq Gold (Life Technologies) and gene-specific primers. We separated PCR products using agarose gels, labeled DNA with 1× SYBR I green nucleic acid gel stain (Life Technologies product number S7567)/1× TBE for 1 h, and quantified band intensities using a transilluminator, CCD camera, XcitaBlue™ conversion screen, and Image Lab image acquisition and analysis software (Image Lab version 5.2.1; Bio-Rad), as previously described[59]. Sequences for primers targeting alternative splicing of *Atp2a1*, *Clcn1*, *Clasp1*, and *Titin*, are shown in Supplementary Table 2.

## Quantification of alternative splicing and gene expression by droplet digital PCR (ddPCR)

To quantify alternative splicing of Clcn1 exon 7a, we used Primer3 software[60,61] to design a Fam-labeled assay targeting the exon 7a–exon 7 splice site and a separate Hex-labeled assay targeting the exon 6–exon 7 splice site. We used ddPCR Supermix for probes (Bio-Rad product number 186-3010), an automated droplet generator (Bio-Rad QX200), automated droplet reader (Bio-Rad QX200), and PCR cycling conditions according to manufacturer instructions, as previously described[59]. For ddPCR quantification of myosin heavy chain gene expression, we used commercially available Fam-labeled primer probe assays for *Myh2* (Type 2 A; Bio-Rad unique assay ID qMmuCEP0055637; amplicon length 88 bp), *Myh1* (Type 2X; Bio-Rad unique assay ID qMmuCIP0033700; amplicon length 113 bp), *Myh4* (Type 2B; Bio-Rad unique assay ID qMmuCEP0058927; amplicon length 74 bp), and *Myh3* (developmental myosin; Bio-Rad unique assay ID qMmuCIP0034632; amplicon length 186 bp). To measure *ACTA1* transgene expression level, we used a previously published Fam-labeled assays for human *ACTA1* mRNA targeting the exon 1–exon 2 splice site[6] and human *ACTA1* pre-mRNA targeting the intron 1–exon 2 splice site[25]. The Fam-labeled assay that we used to measure mouse *Dmpk* was published previously[38]. For normalization controls, we used commercially available standard assays for mouse *Acta1* (Hex-labeled probe; Bio-Rad unique assay ID qMmuCEP0027908; amplicon length 117 bp). After PCR was complete, the plate was loaded into the droplet reader, processed and analyzed using QuantaSoft software (version 1.7.4; Bio-Rad), and total events quantitated using the mean copy number per microliter of duplicate 20 microliter assays from individual samples, as previously described[59]. The sequence for all custom-designed primers and probes is shown in Supplementary Table 3.

## Sample size

The response to Clcn1 ASO treatment in the LR41;*Mbnl1*−/− mice was unknown. Therefore, we were unable to choose a sample size ahead of time to ensure adequate power to measure pharmacodynamic activity. Instead we estimated sample sizes based on RT-PCR analysis of Clcn1

ASO exon skipping patterns in a prior study[24]. Mice ranged from 7 weeks to 18 months of age and were chosen randomly by genotype and stratified by sex to allow an approximately equal number of females and males. Although one or two examiners were blinded to treatment assignments for the ASO studies, the splicing analysis and MyHC imaging quantification data were so robust that it effectively identified the invert control oligo treatment groups from the Clcn1 ASO treatment groups prior to unblinding. Due to the obvious phenotype differences between LR41;*Mbnl1*−/− and LR41, it was impossible to blind for the activity monitoring studies. To verify reproducibility of experimental findings, we used three DM1 mouse models overall, two DM1 mouse models for ASO experiments, two time points, and two methods to quantify Clcn1 alternative splicing patterns.

## Statistical analysis

For two-group and multi-group comparisons, we used unpaired two-tailed *t*-test or analysis of variance (ANOVA), respectively (Prism software version 9.4.1; GraphPad, Inc.). Group data are presented as mean ± s.e.m. A *P*-value <0.05 was considered significant.

## Reporting summary

Further information on research design is available in the Nature Portfolio Reporting Summary linked to this article.

## Data availability

Source data are provided with this paper.

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

## Acknowledgements
We thank Drs. F. Rigo and C.F. Bennett for providing the *ACTA1*-targeting ASO used in the study, and the Elaine and Richard Slye Fund (T.M.W.) and the Muscular Dystrophy Association (award ID 234905; T.M.W.) for support.

## Author contributions
N.H., E.K., L.A., and T.M.W. performed experiments and analyzed data. T.M.W. designed the study and wrote the paper.

## Competing interests
The authors declare no competing interests.
