## [Peer Review File · Nature Communications]

Correction of Clcn1 alternative splicing reverses muscle fiber type transition in mice with myotonic dystrophyREVIEWER COMMENTS

Reviewer #1 (Remarks to the Author):

A prominent clinical feature of myotonic dystrophy type 1 (DM1) is myotonia. This feature is caused by loss of function of the CLCN1 muscle chloride channel due to the expression of CUG expansion RNAs and inhibition of MBNL splicing factors leading to CLCN1 mRNA nonsense-mediated decay (NMD). In this report, Hu et al address the important question of a potential role of myotonia in muscle fiber type glycolytic to oxidative fiber transitions in DM1. They crossed HSALR41 transgenics expressing 160 to 215 CUG repeats to Mbnl1-/- knockouts to generate HSALR41; Mbnl1-/- double homozygous mice. These mice showed both decreased activity profiles and increases in MyHC 1 and 2A oxidative fibers concurrent with a decrease in glycolytic MyHC 2B and X. Following treatment with an ASO that targets Clcn1 exon 7a, the splicing of this cryptic exon was reduced and MyHC 2A frequency decreased in HSALR41; Mbnl1-/- muscles. Overall, this study clarifies a role for myotonia in fiber type switching, demonstrates that this switch is reversible by correction of Clcn1 mis-splicing and highlights possible clinical benefits for this type of ASO-based therapy. However, there are several experimental issues that should be addressed.

Major:

1. Results, Fig. 1a and ln 99-100. The authors describe that their goal is "To test the hypothesis that CUGexp RNA induces pathogenic effects in muscle tissue that are independent of MBNL1 loss-of-function...", so the other model comparison that should be performed in Fig. 1a is Mbnl1-/- alone.
2. Fig. 1 and ln 139-140. The experiment to determine myotonia severity ranking (mild to severe) is not included and should be validated by EMG for these mouse lines.
3. Fig. 3d and ln 208-209. The authors state "In wild type controls, the sarcolemmal signal for CIC-1 appeared more intense in MyHC type 2A and 2X fibers than in 2B fibers." This is not apparent from the image shown.
4. Fig. 5 and ln 245-246. Given the presence of hybrid fibers, this brings the previous analysis of fiber type into question where the authors did not actually stain for Type 2X fibers. The authors need to comment on whether this new information influences their previous results and conclusions.
5. Fig. 7 and ln 262-263. Provide a rationale for testing ACTA1-CUGexp levels following Clcn1 splicing correction and include controls, in addition to Dmpk, for other RNAs encoding sarcomeric proteins to determine if the observed transcriptional increases are specific to ACTA1/Acta1.

Minor:

6. Abstract, ln26. The authors should briefly discuss the relationship between type 1 fiber atrophy and increased frequency of oxidative fiber in DM1.
7. Fig. S1d,e. These figures are mis-labeled a,c. and a sentence is repeated from the beginning of Fig. S1.
8. Results, ln 90-92. The authors state that the LR41 mouse develops mild myotonia (is this only by EMG analysis?) yet when pinch tested (ln 104-105), the authors state this model fails to show myotonia. Please clarify.
9. ln 107-109. Fig. S2 (localization of CUGexp RNA and MBNL2) is mis-placed in this description of spontaneous activity - Fig. S1d is correct for this description.
10. ln 123-124. Comment on the relevance of the two CTG repeat lengths in HSALR41?

11. Ln 151. Provide a justification for using old LR20b mice for this experiment.

Reviewer #2 (Remarks to the Author):

Myotonic dystrophy type 1, the most common adult-onset muscle dystrophy, is caused by large expansions of a CTG microsatellite, located in the 3'untranslated region of the DMPK gene. These CTG repeats are transcribed into toxic RNAs that sequester the MBNL-1 and -2 RNA binding proteins, resulting in multiple mRNA metabolism changes, including a splicing alteration of the chloride channel (Clcn1) mRNA, resulting in skeletal muscle hyperexcitability (myotonia). Despite major progresses in deciphering the molecular mechanisms at play in DM1, some key physio-pathological questions remained unanswered, notably the cause of the progressive loss of glycolytic fibers versus predominance of oxidative skeletal muscle fibers was unclear.

In this work, Hu and collaborators developed a novel animal model of DM1 by crossing mice expressing CUG repeats under the HSA promoter (HAS-LR) with Mbnl1 knockout animals. These mice present a more severe muscle phenotype than the sole HAS-LR or MBNL1 KO mice, and importantly reproduce the typical DM1 switch of glycolytic vs oxidative fibers, associated to stronger splicing changes, notably of Clcn1. Then, using an antisense oligonucleotide correcting Clcn1 splicing, the authors elegantly demonstrated that loss of the muscle chloride channel and the resulting muscle hyperexcitability was most likely causing the glycolytic toward oxidative switch of muscle fiber type. This is important as these data support that muscle fiber switch is reversible in DM1, opening therapeutical options for this disease. Furthermore, these data rule out the hypothesis of specific loss and wasting of glycolytic fibers in myotonic dystrophy.

Overall, Hu et al., data are solid and clear, experiments are well thought and technically well controlled and the final results are novel and of general interest for people working on skeletal muscle physiology and/or on DM, thus supporting publication in Nature Communication.

Minor revisions:

- Extend of myotonia in the different mouse models studied by the authors would be best measured/ confirmed by EMG (cf. Maury Swanson work on Mbnl1 KO mice in Lee et al., 2013 and of course Wheeler et al., 2012; etc. on HAS-LR mice).**
- It would be exciting to investigate by RNA sequencing how strong are the splicing alterations (notably whether DMD, BIN1, CACNA1S, etc. are also altered) in HAS-LR x Mbnl1 Ko mice, compared to the published HAS-LR, Mbnl-1 simple KO and MBNL-1&-2 double KO animal models.**
- Eventually, localization of CLCN1 on T-tubules vs plasma membrane would be best studies on isolated muscle fiber (with RYR, CACNA1S or BIN1 as control?).**
- Introduction, lane 53, MBNL are required for multiple mRNA metabolism regulations (polyadenylation, etc.) and not solely for alternative splicing. Also, DM is likely caused by a progressive titration and thus decrease progressive functions of MBNL rather than a complete "loss".**
- The advantage and risk of anti-myotonic drugs (mexiletine) vs ASO strategy could be discussed more extensively, notably the adverse effects of mexiletine in DM1 with cardiac conduction defects (sodium conductance, splicing change of SCN5A, etc.), but also the potential toxicity of the current ASO chemistry (as shown by the arrested clinical trials on Htt-polyQ & Huntington, C9ORF72 & ALS, DNM2 & CNM, etc.).**

Hu, et al.,

Response to Reviewer comments

REVIEWER COMMENTS

Reviewer #1 (Remarks to the Author):

A prominent clinical feature of myotonic dystrophy type 1 (DM1) is myotonia. This feature is caused by loss of function of the CLCN1 muscle chloride channel due to the expression of CUG expansion RNAs and inhibition of MBNL splicing factors leading to CLCN1 mRNA nonsense-mediated decay (NMD). In this report, Hu et al address the important question of a potential role of myotonia in muscle fiber type glycolytic to oxidative fiber transitions in DM1. They crossed HSALR41 transgenics expressing 160 to 215 CUG repeats to Mbnl1^{-/-} knockouts to generate HSALR41; Mbnl1^{-/-} double homozygous mice. These mice showed both decreased activity profiles and increases in MyHC 1 and 2A oxidative fibers concurrent with a decrease in glycolytic MyHC 2B and X. Following treatment with an ASO that targets Clcn1 exon 7a, the splicing of this cryptic exon was reduced and MyHC 2A frequency decreased in HSALR41; Mbnl1^{-/-} muscles. Overall, this study clarifies a role for myotonia in fiber type switching, demonstrates that this switch is reversible by correction of Clcn1 mis-splicing and highlights possible clinical benefits for this type of ASO-based therapy. However, there are several experimental issues that should be addressed.

Major:

"1. Results, Fig. 1a and ln 99-100. The authors describe that their goal is "To test the hypothesis that CUGexp RNA induces pathogenic effects in muscle tissue that are independent of MBNL1 loss-of-function", so the other model comparison that should be performed in Fig. 1a is Mbnl1^{-/-} alone."

Response: We added data from the Mbnl1^{-/-} alone model to Fig. 1a and Supplementary Fig.1. It shows that Mbnl1^{-/-} mice appear similar to age-matched LR41 and significantly different from age-matched LR41;Mbnl1^{-/-} in terms of vertical activity, intermediate between LR41 and

LR41;Mbnl1^{-/-} for distance traveled, rest time, and ambulatory time, and similar to LR41;Mbnl1^{-/-} for stereotypic time.

"2. Fig. 1 and ln 139-140. The experiment to determine myotonia severity ranking (mild to severe) is not included and should be validated by EMG for these mouse lines."

Response: We agree that EMG analysis of mice would be ideal. However, we have never had EMG available for use in mice. We have added a new Supplementary Video 2 that shows visible myotonia in the LR20b model, demonstrating that it is less severe and of shorter duration than in the LR41;Mbnl1^{-/-} model shown in Supplementary Video 1.

Mis-regulated alternative splicing of Clcn1 transcripts, leading to increased inclusion of frame-shifting exon 7a and a resulting premature termination codon, is the cause of myotonia in DM1 patients and DM1 mouse models (Mankodi, et al., 2002; Charlet, et al., 2002; Wheeler, et al., 2007). Immunofluorescence analysis of muscle tissue sections has revealed that CIC-1 protein localization at the membrane appears graded by genotype: *Mbnl1*^{-/-} has less than LR20b, which has less than LR41 (Mankodi, et al., 2002; Wheeler, et al., 2007). The disease mechanism involving titration of Mbnl1 protein by CUG^{exp} RNA (as Reviewer 2 indicates) leads to partial loss of Mbnl1 splicing regulation in the LR41 and LR20b lines, and complete loss of Mbnl1 splicing regulation in the *Mbnl1*^{-/-} model. In the tibialis anterior (TA) muscle, Clcn1 exon 7a inclusion is about 37% in LR20 and 60% in *Mbnl1*^{-/-} (Wheeler, et al., 2007), while in the gastrocnemius muscle, Clcn1 exon 7a inclusion is about 50% in LR20 and 60% in *Mbnl1*^{-/-} (Hu, et al., 2021). The greater degree of Clcn1 mis-splicing in *Mbnl1*^{-/-} vs. LR20b accounts for the more severe myotonia in *Mbnl1*^{-/-} than in LR20b. The greater degree of mis-regulated splicing in LR41;Mbnl1^{-/-} vs *Mbnl1*^{-/-} alone most likely is due to sequestration of MBNL2 protein by CUG^{exp} RNA (shown in Supplementary Fig. 2), leading to compound loss of MBNL protein similar to that observed in genetic models that are homozygous for deletion of Mbnl1 and heterozygous for deletion of Mbnl2 (*Mbnl1*^{-/-};*Mbnl2*^{+/-} in Lee, et al., 2013; or *Mbnl3/4* KO in Tanner, et al., 2021).

- Charlet, B.N., Savkur, R.S., Singh, G., Philips, A.V., Grice, E.A. & Cooper, T.A. Loss of the muscle-specific chloride channel in type 1 myotonic dystrophy due to misregulated alternative splicing. *Mol Cell* **10**, 45-53 (2002).

- Hu, N., Kim, E., Antoury, L., Li, J., Gonzalez-Perez, P., Rutkove, S.B. & Wheeler, T.M. Antisense oligonucleotide and adjuvant exercise therapy reverse fatigue in old mice with myotonic dystrophy. *Mol Ther Nucleic Acids* **23**, 393-405 (2021).
- Lee, K.Y., Li, M., Manchanda, M., Batra, R., Charizanis, K., Mohan, A., Warren, S.A., Chamberlain, C.M., Finn, D., Hong, H., Ashraf, H., Kasahara, H., Ranum, L.P. & Swanson, M.S. Compound loss of muscleblind-like function in myotonic dystrophy. *EMBO Mol Med* **5**, 1887-1900 (2013).
- Mankodi, A., Logigian, E., Callahan, L., McClain, C., White, R., Henderson, D., Krym, M. & Thornton, C.A. Myotonic dystrophy in transgenic mice expressing an expanded CUG repeat. *Science* **289**, 1769-1773 (2000).
- Mankodi, A., Takahashi, M.P., Jiang, H., Beck, C.L., Bowers, W.J., Moxley, R.T., Cannon, S.C. & Thornton, C.A. Expanded CUG repeats trigger aberrant splicing of CIC-1 chloride channel pre-mRNA and hyperexcitability of skeletal muscle in myotonic dystrophy. *Mol Cell* **10**, 35-44 (2002).
- Tanner, M.K., Tang, Z. & Thornton, C.A. Targeted splice sequencing reveals RNA toxicity and therapeutic response in myotonic dystrophy. *Nucleic Acids Res* **49**, 2240-2254 (2021).
- Wheeler, T.M., Lueck, J.D., Swanson, M.S., Dirksen, R.T. & Thornton, C.A. Correction of CIC-1 splicing eliminates chloride channelopathy and myotonia in mouse models of myotonic dystrophy. *J Clin Invest* **117**, 3952-3957 (2007).

"3. Fig. 3d and ln 208-209. The authors state "In wild type controls, the sarcolemmal signal for CIC-1 appeared more intense in MyHC type 2A and 2X fibers than in 2B fibers." This is not apparent from the image shown."

Response: In WT muscle, the lower signal at the membrane of 2B fibers as compared to 2A or 2X is most evident when 2B fibers are adjacent to other 2B fibers. To improve visualization of this point, we have created a new Fig. 4 that features larger images and asterisks (*) to highlight WT 2B fibers adjacent to other 2B fibers and show a lower CIC-1 protein intensity at the membrane as compared to 2A fibers, which are indicated by the "+" symbol.

"4. Fig. 5 and ln 245-246. Given the presence of hybrid fibers, this brings the previous analysis of fiber type into question where the authors did not actually stain for Type 2X fibers. The authors need to comment on whether this new information influences their previous results and conclusions."

Response: We are unable to co-label with the 2X and 2B antibodies because both are isotype IgM. Therefore, initially we labeled muscles using antibodies to Type 1, 2A, and 2B, leaving the 2X fibers unlabeled. MyHC 2A and 2B were identified as fibers with > 50% intensity of the brightest, and 2X as fibers showing < 50% intensity. This revealed a large overall shift from 2A to 2B, which probably progresses through the following pattern: 2A > 2A/2X hybrid > 2X > 2X/2B hybrid > 2B (Schiaffino and Reggiani, 2011). The variable intensity of MyHC 2A and 2B labeling suggested the presence of hybrid fibers (Figs. 1b, 3a, and 4). The subsequent ddPCR results suggested that we were overestimating the population of pure 2A fibers, prompting co-labeling with antibodies against MyHC 2A and 2X. This enabled a more precise quantification of 2A, 2A/2X hybrid, and 2X fibers, but our conclusion of an overall shift from 2A to 2B remained unchanged.

- Schiaffino, S. & Reggiani, C. Fiber types in mammalian skeletal muscles. *Physiol Rev* **91**, 1447-1531 (2011).

"5. Fig. 7 and ln 262-263. Provide a rationale for testing ACTA1-CUG^{exp} levels following *Clcn1* splicing correction and include controls, in addition to *Dmpk*, for other RNAs encoding sarcomeric proteins to determine if the observed transcriptional increases are specific to ACTA1/*Acta1*."

Response: We've added the following text to the Results section, "In prior studies, alternative splicing of *Atp2a1* served as a sensitive indicator of ACTA1-CUG^{exp} transcript levels and therapeutic drug activity (Wheeler, et al., 2012; Hu, et al., 2018). Having shown that selective correction of *Clcn1* splicing exacerbated mis-regulated splicing of *Atp2a1* in LR20b mice, we next examined the effect of *Clcn1* splicing correction on expression of ACTA1-CUG^{exp} transcripts in treated muscles."

We also modified the Discussion, as follows: "The greater abundance of *ACTA1*-CUG^{exp} transcripts that we observed in *Clcn1* ASO-treated muscles could be interpreted as cause for concern in the translation of this therapeutic approach to DM1 patients, perhaps by increasing the production and/or enhancing the stability of *DMPK*-CUG^{exp} RNA in skeletal muscle. This is unlikely for several reasons, 1) the elevation of *ACTA1*-CUG^{exp} pre-mRNA and mRNA to a similar degree in ASO-treated muscles suggests that increased transcription of the *ACTA1* transgene is the mechanism, 2) the strong negative correlation of *ACTA1*-CUG^{exp} transcript levels with *Myh2* (2A) and *Myh1* (2X) transcripts, and strong positive correlation with *Myh4* (2B) suggest that expression of the human *ACTA1* transgene is higher in 2B fibers and that the increase of *ACTA1*-CUG^{exp} RNA observed in ASO-treated muscles was an artifact of the human *ACTA1* transgene responding to the fiber type shift, and 3) transcript levels of *Dmpk*, the gene that carries the expanded repeat in human DM1, appeared unchanged in treated muscles, arguing against a similar *Clcn1*-targeting ASO leading to increased *DMPK*-CUG^{exp} expression. In fact, the therapeutic effects that we observed in the setting of higher *ACTA1*-CUG^{exp} RNA content suggests the potential for an even greater clinical benefit of fiber type transition in DM1 patients if *DMPK*-CUG^{exp} expression remains stable."

- Hu, N., Antoury, L., Baran, T.M., Mitra, S., Bennett, C.F., Rigo, F., Foster, T.H. & Wheeler, T.M. Non-invasive monitoring of alternative splicing outcomes to identify candidate therapies for myotonic dystrophy type 1. *Nat Commun* **9**, 5227 (2018).
- Wheeler, T.M., Leger, A.J., Pandey, S.K., MacLeod, A.R., Nakamori, M., Cheng, S.H., Wentworth, B.M., Bennett, C.F. & Thornton, C.A. Targeting nuclear RNA for in vivo correction of myotonic dystrophy. *Nature* **488**, 111-115 (2012).

Minor:

"6. Abstract, ln26. The authors should briefly discuss the relationship between type 1 fiber atrophy and increased frequency of oxidative fiber in DM1."

Response: The abstract is already at the 150-word limit. Therefore, we added the following text to the Introduction in **bold**, "In cross sectional studies of human DM, the degree of muscle weakness is associated with **Type 1 oxidative fiber atrophy** and a greater proportion of

mechanically less powerful oxidative fibers, which may reflect fiber type transition or a preferential loss of glycolytic fibers (Borg, et al., 1987; Tohgi, et al., 1994)."

- Borg, J., Edstrom, L., Butler-Browne, G.S. & Thornell, L.E. Muscle fibre type composition, motoneuron firing properties, axonal conduction velocity and refractory period for foot extensor motor units in dystrophia myotonica. *J Neurol Neurosurg Psychiatry* **50**, 1036-1044 (1987).
- Tohgi, H., Kawamorita, A., Utsugisawa, K., Yamagata, M. & Sano, M. Muscle histopathology in myotonic dystrophy in relation to age and muscular weakness. *Muscle Nerve* **17**, 1037-1043 (1994).

"7. Fig. S1d,e. These figures are mis-labeled a,c. and a sentence is repeated from the beginning of Fig. S1."

Response: We corrected these errors.

"8. Results, ln 90-92. The authors state that the LR41 mouse develops mild myotonia (is this only by EMG analysis?), yet when pinch tested (ln 104-105), the authors state this model fails to show myotonia. Please clarify."

Response: In the first paragraph of the Results section, we have added the following text in Bold, "HSA^{LR} line LR41 features a lower transgene copy number, a lower *ACTA1*-CUG^{exp} RNA abundance, mis-regulated alternative splicing to a lesser degree, **less frequent myotonia (40% of mice examined by electromyography/EMG)**, and milder myopathy than HSA^{LR} line LR20b (Mankodi, et al., 2000 and 2002)."

In the Supplementary Video 1 legend for the current manuscript, we have added the following text in **bold**, "By contrast, grasping an age-matched homozygous LR41 littermate at the base of the tail elicits no **visible** myotonia, **although some of these mice feature electrical myotonia detectable by needle electromyography (EMG) examination (Mankodi, et al., 2000)**."

- Mankodi, A., Logigian, E., Callahan, L., McClain, C., White, R., Henderson, D., Krym, M. & Thornton, C.A. Myotonic dystrophy in transgenic mice expressing an expanded CUG repeat. *Science* **289**, 1769-1773 (2000).
- Mankodi, A., Takahashi, M.P., Jiang, H., Beck, C.L., Bowers, W.J., Moxley, R.T., Cannon, S.C. & Thornton, C.A. Expanded CUG repeats trigger aberrant splicing of CIC-1 chloride channel pre-mRNA and hyperexcitability of skeletal muscle in myotonic dystrophy. *Mol Cell* **10**, 35-44 (2002).

"9. Ln 107-109. Fig. S2 (localization of CUG^{exp} RNA and MBNL2) is mis-placed in this description of spontaneous activity - Fig. S1d is correct for this description."

Response: We corrected this error.

"10. Ln 123-124. Comment on the relevance of the two CTG repeat lengths in HSALR41?"

Response: The two CTG repeat lengths in LR41 suggests the possibility of two transgene copies, and that the CTG repeat length in one of them has contracted more than in the other.

"11. Ln 151. Provide a justification for using old LR2ob mice for this experiment."

Response: We've added the following text to the main file, "In a prior study we found that treadmill-walking exercise combined with ACTA1-CUG^{exp} transcript-targeting ASO treatment for 3 months rescued fatigue in old LR20b mice (Hu, et al., 2021). Chronic exercise has been associated with muscle fiber type transition (Schiaffino and Reggiani, 2011). For example, the percentage of type 2A fibers was higher and 2B fibers lower in mice engaged in voluntary wheel running for four weeks as compared to mice that received no exercise (Allen, et al., 2001)."

- Allen, D.L., Harrison, B.C., Maass, A., Bell, M.L., Byrnes, W.C. & Leinwand, L.A. Cardiac and skeletal muscle adaptations to voluntary wheel running in the mouse. *J Appl Physiol* (1985) **90**, 1900-1908 (2001).

- Hu, N., Kim, E., Antoury, L., Li, J., Gonzalez-Perez, P., Rutkove, S.B. & Wheeler, T.M. Antisense oligonucleotide and adjuvant exercise therapy reverse fatigue in old mice with myotonic dystrophy. *Mol Ther Nucleic Acids* **23**, 393-405 (2021).
- Schiaffino, S. & Reggiani, C. Fiber types in mammalian skeletal muscles. *Physiol Rev* **91**, 1447-1531 (2011).

Reviewer #2 (Remarks to the Author):

Myotonic dystrophy type 1, the most common adult-onset muscle dystrophy, is caused by large expansions of a CTG microsatellite, located in the 3' untranslated region of the DMPK gene. These CTG repeats are transcribed into toxic RNAs that sequester the MBNL-1 and -2 RNA binding proteins, resulting in multiple mRNA metabolism changes, including a splicing alteration of the chloride channel (Clcn1) mRNA, resulting in skeletal muscle hyperexcitability (myotonia). Despite major progresses in deciphering the molecular mechanisms at play in DM1, some key physio-pathological questions remained unanswered, notably the cause of the progressive loss of glycolytic fibers versus predominance of oxidative skeletal muscle fibers was unclear.

In this work, Hu and collaborators developed a novel animal model of DM1 by crossing mice expressing CUG repeats under the HSA promoter (HAS-LR) with Mbnl1 knockout animals. These mice present a more severe muscle phenotype than the sole HAS-LR or MBNL1 KO mice, and importantly reproduce the typical DM1 switch of glycolytic vs oxidative fibers, associated to stronger splicing changes, notably of Clcn1. Then, using an antisense oligonucleotide correcting Clcn1 splicing, the authors elegantly demonstrated that loss of the muscle chloride channel and the resulting muscle hyperexcitability was most likely causing the glycolytic toward oxidative switch of muscle fiber type. This is important as these data support that muscle fiber switch is reversible in DM1, opening therapeutical options for this disease. Furthermore, these data rule out the hypothesis of specific loss and wasting of glycolytic fibers in myotonic dystrophy.

Overall, Hu et al., data are solid and clear, experiments are well thought and technically well controlled and the final results are novel and of general interest for people working on skeletal muscle physiology and/or on DM, thus supporting publication in Nature Communication.

Minor revisions:

"- Extend of myotonia in the different mouse models studied by the authors would be best measured/ confirmed by EMG (cf. Maury Swanson work on Mbnl1 KO mice in Lee et al., 2013 and of course Wheeler et al., 2012; etc. on HSA-LR mice)."

Response: We agree, but have never had EMG available for our use in mice. Please see the response to Point 2 by Reviewer 1.

"- It would be exciting to investigate by RNA sequencing how strong are the splicing alterations (notably whether DMD, BIN1, CACNA1S, etc. are also altered) in HSA-LR x Mbnl1 Ko mice, compared to the published HSA-LR, Mbnl-1 simple KO and MBNL-1&-2 double KO animal models."

Response: We have performed RNA sequencing in the LR41;Mbnl1^{-/-} model as part of a separate study. Alternative splicing of *Cacna1s* is altered in this model and we have added these data to Fig. 2. We ended up following *Atp2a1*, *Clasp1*, and *Ttn* alternative splicing throughout the study because alternative exon inclusion for these three transcripts in LR20b is intermediate between LR41;Mbnl1^{-/-} and WT, whereas alternative exon inclusion of *Cacna1s* in LR20b appears similar to WT.

"- Eventually, localization of CLCN1 on T-tubules vs plasma membrane would be best studies on isolated muscle fiber (with RYR, CACNA1S or BIN1 as control?)."

Response: We agree. One limitation could be finding the right antibody. In our experience, that anti-CLC-1 antibody that we used in this study works well on unfixed cryosections, but poorly on fixed tissue.

- "Introduction, line 53, MBNL are required for multiple mRNA metabolism regulations (polyadenylation, etc.) and not solely for alternative splicing. Also, DM is likely caused by a progressive titration and thus decrease progressive functions of MBNL rather than a complete "loss"."

Response: We modified the Introduction text as follows, "This pathogenic RNA readily binds proteins in the muscleblind-like (MBNL) family that are required for normal regulation of alternative splicing, gene expression, transcript stability, and alternative polyadenylation, resulting in partial loss of MBNL protein function (Miller, et al., 2000; Lin et al., 2006; Du, et al., 2010; Batra, et al., 2014; Wang, et al., 2015)."

- Batra, R., Charizanis, K., Manchanda, M., Mohan, A., Li, M., Finn, D.J., Goodwin, M., Zhang, C., Sobczak, K., Thornton, C.A. & Swanson, M.S. Loss of MBNL leads to disruption of developmentally regulated alternative polyadenylation in RNA-mediated disease. *Mol Cell* **56**, 311-322 (2014).
- Du, H., Cline, M.S., Osborne, R.J., Tuttle, D.L., Clark, T.A., Donohue, J.P., Hall, M.P., Shiue, L., Swanson, M.S., Thornton, C.A. & Ares, M., Jr. Aberrant alternative splicing and extracellular matrix gene expression in mouse models of myotonic dystrophy. *Nat Struct Mol Biol* **17**, 187-193 (2010).
- Lin, X., Miller, J.W., Mankodi, A., Kanadia, R.N., Yuan, Y., Moxley, R.T., Swanson, M.S. & Thornton, C.A. Failure of MBNL1-dependent post-natal splicing transitions in myotonic dystrophy. *Hum Mol Genet* **15**, 2087-2097 (2006).
- Miller, J.W., Urbinati, C.R., Teng-Umnuay, P., Stenberg, M.G., Byrne, B.J., Thornton, C.A. & Swanson, M.S. Recruitment of human muscleblind proteins to (CUG)(n) expansions associated with myotonic dystrophy. *Embo J* **19**, 4439-4448 (2000).
- Wang, E.T., Ward, A.J., Cherone, J.M., Giudice, J., Wang, T.T., Treacy, D.J., Lambert, N.J., Freese, P., Saxena, T., Cooper, T.A. & Burge, C.B. Antagonistic regulation of mRNA expression and splicing by CELF and MBNL proteins. *Genome Res* **25**, 858-871 (2015).

- "The advantage and risk of anti-myotonic drugs (mexiletine) vs ASO strategy could be discussed more extensively, notably the adverse effects of mexiletine in DM1 with cardiac conduction defects (sodium conductance, splicing change of *SCN5A*, etc.), but also the potential toxicity of the current ASO chemistry (as shown by the arrested clinical trials on *Htt-polyQ* & Huntington, *C9ORF72* & ALS, *DNM2* & *CNM*, etc.)."

Response: Regarding mexiletine, we've updated the relevant portion of the Discussion as follows, "Another disadvantage of mexiletine is that it has pro-arrhythmic effects on the heart, which complicates its use in individuals with DM1 who already are prone to cardiac conduction disturbance (McNally, et al., 2020)."

Regarding potential ASO toxicity, we have updated the relevant portion of the Discussion as follows, "In addition, an early phase clinical trial of an RNase H-active ASO for the treatment of myotubular myopathy recently was terminated due to poor tolerability at the low dose (clinicaltrials.gov identifier NCT04033159), suggesting that translation of a cleavage-based approach to muscle may be difficult. By contrast, the splice-shifting morpholino chemistry that we have used in this study has demonstrated safety in humans, with four drugs already FDA-approved for the treatment of Duchenne muscular dystrophy (Holm, et al., 2022)."

- Holm, A., Hansen, S.N., Klitgaard, H. & Kauppinen, S. Clinical advances of RNA therapeutics for treatment of neurological and neuromuscular diseases. *RNA Biol* **19**, 594-608 (2022).
- McNally, E.M., Mann, D.L., Pinto, Y., Bhakta, D., Tomaselli, G., Nazarian, S., Groh, W.J., Tamura, T., Duboc, D., Itoh, H., Hellerstein, L. & Mammen, P.P.A. Clinical Care Recommendations for Cardiologists Treating Adults With Myotonic Dystrophy. *J Am Heart Assoc* **9**, e014006 (2020).

REVIEWERS' COMMENTS

Reviewer #1 (Remarks to the Author):

The authors have addressed all of my prior concerns.

Reviewer #2 (Remarks to the Author):

The authors adequately respond to my comments, hence I would suggest that this work is suitable for publication in Nat Comm

Hu, et al.,

Response to Reviewer comments

REVIEWER COMMENTS

Reviewer #1 (Remarks to the Author):

The authors have addressed all of my prior concerns.

Response: Thank you.

Reviewer #2 (Remarks to the Author):

The authors adequately respond to my comments, hence I would suggest that this work is suitable for publication in Nat Comm.

Response: Thank you.